# Distinct forms of synaptic plasticity during ascending vs descending control of medial olivocochlear efferent neurons

Gabriel E Romero[1], Laurence O Trussell[2]*

[1]Physiology & Pharmacology Graduate Program, Oregon Health & Science University, Portland, United States; [2]Oregon Hearing Research Center and Vollum Institute, Oregon Health & Science University, Portland, United States

**Abstract** Activity in each brain region is shaped by the convergence of ascending and descending axonal pathways, and the balance and characteristics of these determine the neural output. The medial olivocochlear (MOC) efferent system is part of a reflex arc that critically controls auditory sensitivity. Multiple central pathways contact MOC neurons, raising the question of how a reflex arc could be engaged by diverse inputs. We examined functional properties of synapses onto brainstem MOC neurons from ascending (ventral cochlear nucleus, VCN) and descending (inferior colliculus, IC) sources in mice using an optogenetic approach. We found that these pathways exhibited opposing forms of short-term plasticity, with the VCN input showing depression and the IC input showing marked facilitation. By using a conductance-clamp approach, we found that combinations of facilitating and depressing inputs enabled firing of MOC neurons over a surprisingly wide dynamic range, suggesting an essential role for descending signaling to a brainstem nucleus.

**\*For correspondence:**
trussell@ohsu.edu

**Competing interests:** The authors declare that no competing interests exist.

## Introduction

The cochlea is the peripheral organ of hearing. As such, it communicates with the central nervous system by its centrallyprojecting afferent fibers. However, the cochlea also receives inputs from a population of cochlear efferent fibers that originate in the brainstem. The medial olivocochlear (MOC) system provides many of these efferent fibers, and may serve to protect the cochlea from acoustic trauma (*Rajan, 1988*; *Kujawa and Liberman, 1997*; *Darrow et al., 2007*) and to dynamically enhance the detection of salient sound in diverse sensory environments (*Winslow and Sachs, 1987*; *Kawase and Liberman, 1993*) by controlling cochlear gain in a frequency- and intensity-specific manner. MOC efferent fibers arise from cholinergic neurons whose somata primarily reside in the ventral nucleus of the trapezoid body (VNTB) of the superior olivary complex (SOC) (*Warr, 1975*) and project to outer hair cells in the cochlea (*Guinan et al., 1983*; *Guinan et al., 1984*; *Wilson et al., 1991*), and this peripheral control by efferents has been extensively studied (*Guinan, 2010*; *Guinan, 2018*). MOC fibers respond to sound and form a negative feedback system, and is thus described as a reflex providing frequency-specific feedback to the cochlea (*Liberman and Brown, 1986*; *Winslow and Sachs, 1987*; *Brown, 2016*). This feedback is mediated by acetylcholine released from terminals of MOC fibers, thereby inhibiting outer hair cell motility and decreasing cochlear sensitivity (*Wiederhold and Kiang, 1970*).

In contrast to this detailed understanding of peripheral efferent mechanisms, the electrophysiological properties of the efferent neurons and their control by central pathways remain unclear, and indeed, it is not known even to what extent these neurons function as a reflex arc or as mediators of descending control by higher brain regions. For example, excitatory synaptic inputs that modulate and control MOC neuron function are made both by ascending inputs from the cochlear nucleus

(termed here the 'reflex pathway') and by descending inputs from areas that include brainstem, inferior colliculus (IC), and auditory cortex (*Thompson and Thompson, 1993*; *Vetter et al., 1993*; *Mulders and Robertson, 2002*). The reflex MOC pathway receives ascending auditory inputs from principal neurons in the ventral cochlear nucleus (VCN), possibly by T-stellate cells (*Thompson and Thompson, 1991*; *de Venecia et al., 2005*; *Darrow et al., 2012*; *Brown et al., 2013*). While T-stellate cells are anatomically and physiologically well suited to provide auditory information to MOC neurons (*Oertel et al., 2011*) and receive inputs from type I spiral ganglion neurons, whose axons form the auditory nerve, in fact no direct evidence shows that these neurons activate MOC neurons. Descending projections from the IC contact MOC neurons (*Faye-Lund, 1986*; *Thompson and Thompson, 1993*; *Vetter et al., 1993*) and are tonotopically arranged, as low-frequency fibers project laterally and high frequencies increasingly project more medially (*Caicedo and Herbert, 1993*; *Suthakar and Ryugo, 2017*). Descending inputs may utilize the MOC system to suppress cochlear inputs during non-auditory tasks (*Delano et al., 2007*; *Wittekindt et al., 2014*) and are well positioned to aid sound detection in noise by contextually inhibiting background frequency spectra (*Farhadi et al., 2021*). However, again, direct evidence for the significance of such descending control is lacking and, in particular, whether such inputs can drive the efferent system or merely modify the control mediated by the reflex pathway.

We have investigated the physiological properties of MOC neurons, testing the relative efficacy of synaptic inputs made by reflex vs descending pathways. MOC efferent neurons were labeled for targeted patch-clamp recording in brain slices from 30- to 48-day-old ChAT-IRES-Cre mice, and properties of ascending and descending synaptic inputs onto these neurons from VCN and IC were analyzed using virally driven optogenetic excitation. By making recordings from identified neurons in mature mice, we found that MOC neurons are exceptionally homogeneous in their electrophysiological properties and are well suited to encoding stimulus intensity and duration with sustained firing at constant rates. Synaptic inputs to MOC neurons from the VCN and IC are glutamatergic, and both transmit using fast-gating $Ca^{2+}$-permeable α-amino-3-hydroxy-5-methyl-4-isoxazolepropionic acid (AMPA) receptors. Using a novel intersectional adeno-associated virus (AAV) approach that enabled optical excitation of only T-stellate cells in the VCN, we were able to provide direct evidence that T-stellate cells are an excitatory interneuron involved in MOC reflex circuitry. However, comparing the short-term synaptic plasticity of VCN and IC inputs, we discovered that at the same stimulus rates, VCN inputs exhibited rapid short-term depression, while IC inputs exhibited augmentation, increasing several fold in synaptic strength. Conductance-clamp experiments, in which these inputs were simulated with realistic patterns of activity, showed that descending control of hair cell activity may be a potent means for engaging the full dynamic range of activity of MOC neurons, thus permitting broad control of cochlear sensitivity.

## Results

### Cholinergic auditory efferent neurons are tdTomato-positive in ChAT-Cre/tdTomato mice

The SOC features two groups of cholinergic olivocochlear efferent neurons, lateral olivocochlear (LOC) neurons and MOC neurons (*Warr and Guinan, 1979*). The somata of MOC neurons reside primarily in the VNTB, whereas LOC neurons are smaller, more numerous, and are located in the lateral superior olive (LSO). While MOC neurons exert inhibitory control over outer hair cells in the cochlea, LOC neurons modulate the excitability of the auditory nerve, as they mainly terminate onto dendrites of type I spiral ganglion neurons near sensory inner hair cells (*Liberman, 1980*).

In order to visualize cholinergic efferent neurons in the SOC of acute brain slices for whole-cell recording, we crossed a ChAT-IRES-Cre mouse line with a reporter line, Ai9(RCL-tdT), that expressed the fluorophore tdTomato in a Cre recombinase-dependent manner (*Torres Cadenas et al., 2020*). This cross will be referred to as ChAT-Cre/tdTomato. Neurons positive for tdTomato were visible in the LSO and VNTB of the SOC, and co-labeled with anti-ChAT antibody, confirming they were cholinergic neurons (*Figure 1A–C*). A majority of tdTomato-positive neurons in the ipsilateral LSO and contralateral VNTB were retrogradely labeled by injecting cholera toxin subunit B (CTB) into the cochlea, confirming that they were indeed auditory efferent neurons (*Figure 1D–E*). While MOC neurons project primarily to contralateral cochlea and LOC neurons project primarily to

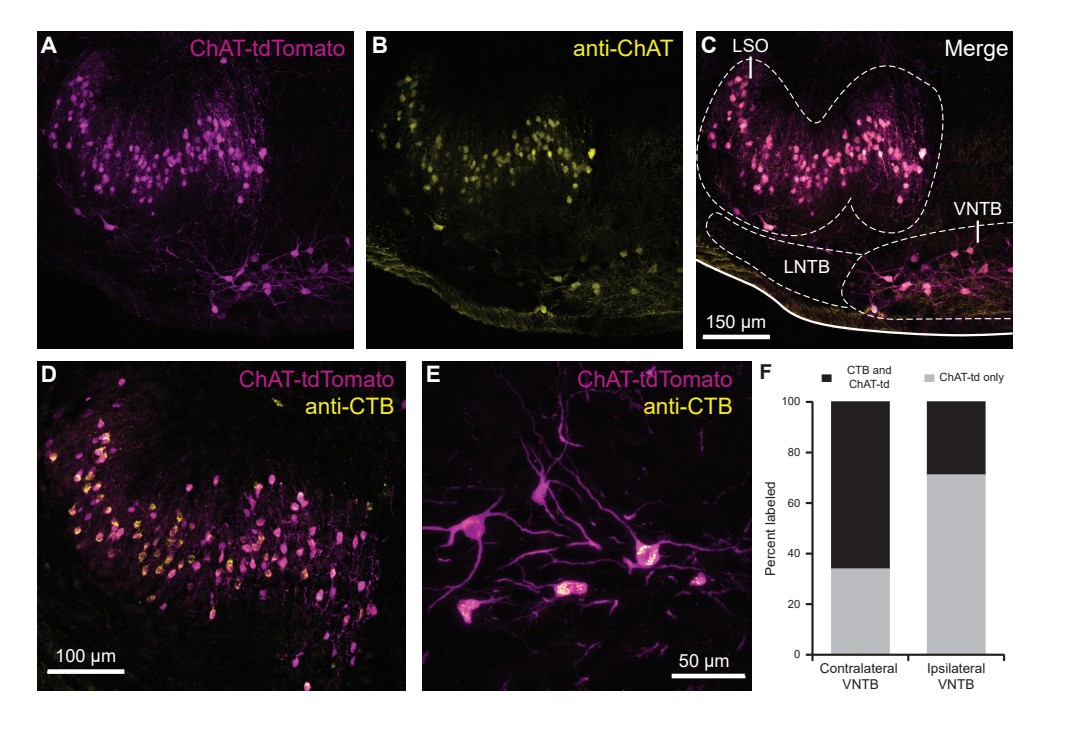

**Figure 1.** Cholinergic auditory efferent neurons identified with the retrograde tracer CTB were tdTomato-positive in ChAT-Cre/tdTomato mice. (**A**) ChAT-Cre/tdTomato-positive neurons in the LSO and VNTB of the superior olivary complex co-labeled with anti-ChAT antibody (**B**) confirming they are cholinergic neurons (**C**). ChAT-Cre/tdTomato-positive neurons in the ipsilateral LSO (**D**) and contralateral VNTB (**E**) were retrogradely labeled by cochlear CTB injections. (**F**) Contralateral to cochlear CTB injections, 66.1% of ChAT-Cre/tdTomato-positive VNTB neurons were labeled. In ipsilateral VNTB, 28.9% of ChAT-Cre/tdTomato=positive neurons were labeled (N = 3 mice, 205 cells). Abbreviations: LNTB, lateral nucleus of the trapezoid body; LSO, lateral superior olive; VNTB, ventral nucleus of the trapezoid body; CTB, cholera toxin subunit B.

ipsilateral cochlea, each group contains fibers projecting to both cochleae (*Warr, 1975*; *Warr and Guinan, 1979*; *Brown and Levine, 2008*). Contralateral to unilateral cochlear CTB injections, 66.1% of tdTomato-positive VNTB neurons were labeled, and in ipsilateral VNTB, 28.9% were labeled (*Figure 1F*).

## Medial olivocochlear neurons accurately encode stimulus intensity and duration

In vivo recordings have revealed that MOC neurons exhibit little or no spontaneous activity and respond to sound in a frequency- and intensity-dependent manner (*Robertson and Gummer, 1985*; *Liberman and Brown, 1986*). To investigate how intrinsic membrane properties of MOC neurons underlie in vivo responses, whole-cell patch-clamp recordings were made from tdTomato-positive MOC neurons in the VNTB from acute brain slices of ChAT-Cre/tdTomato mice. We found that the majority of MOC neurons had a resting membrane potential of $-80.4 \pm 0.8$ mV (N = 56) and were silent at rest (only 3/59 neurons were spontaneously active), consistent with the low frequency of spontaneous activity observed in vivo (*Fex, 1962*; *Cody and Johnstone, 1982*; *Robertson, 1984*; *Robertson and Gummer, 1985*). The membrane capacitance ($C_m$) and resistance ($R_m$) were $36.5 \pm 1.6$ pF and $123 \pm 9$ MΩ (N = 59), respectively. In response to hyperpolarizing current injections, MOC neurons lacked an apparent voltage 'sag', indicating minimal expression of hyperpolarization-activated cyclic nucleotide-gated (HCN) channels (*Figure 2A*). Depolarizing currents near action potential threshold revealed a biphasic after-hyperpolarization waveform following each spike (*Figure 2A*, see arrowhead in +0.5 nA example, observed in 57/59 MOC neurons). In response to the increasing amplitude of 500-ms depolarizing current injections, MOC neurons fired action potentials that encoded stimulus current intensity with a remarkably linear increase in spike rate (*Figure 2A–D*). For injections up to 900 pA, MOC neurons (N = 33) responded with linearly

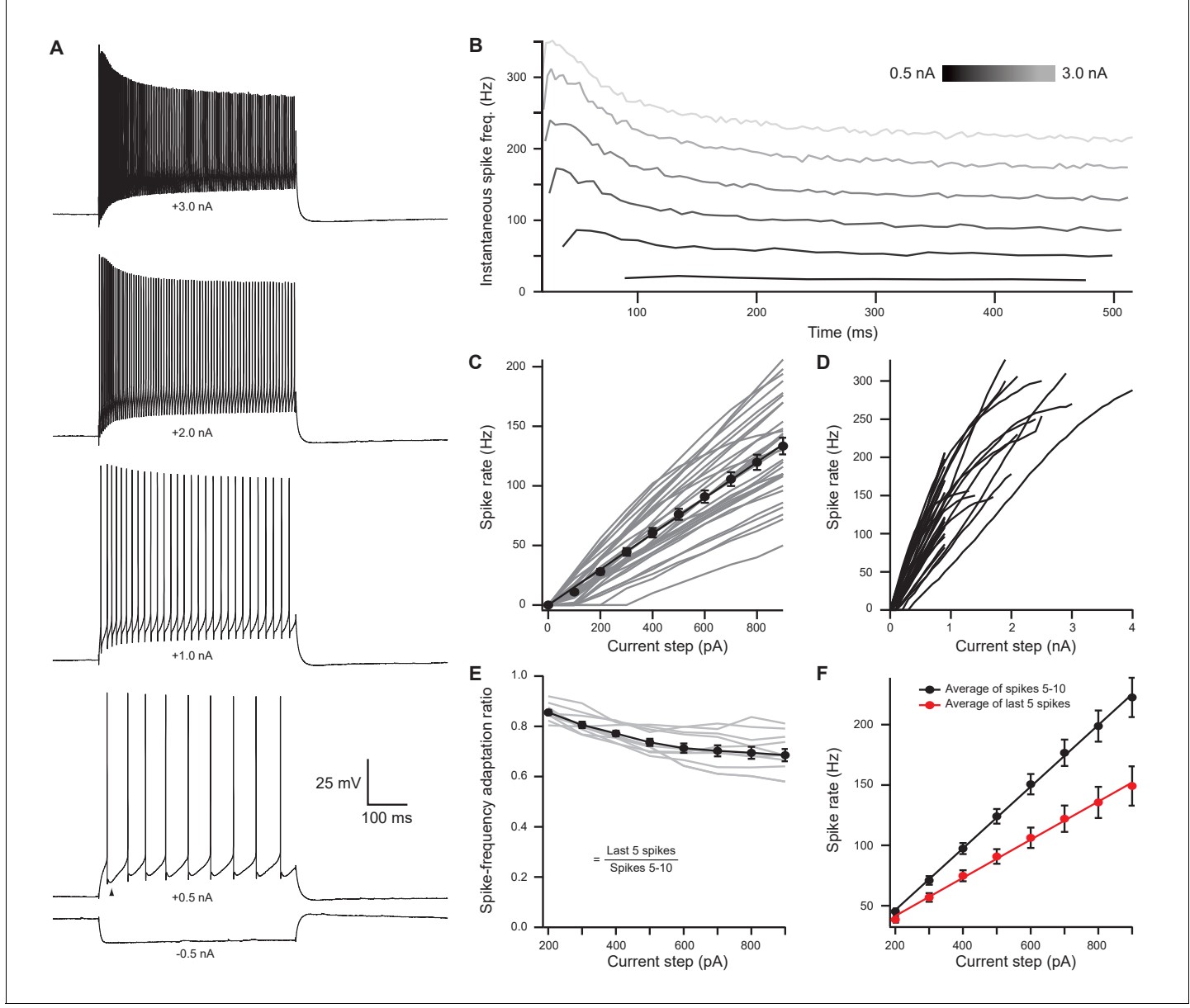

**Figure 2.** Medial olivocochlear neurons accurately encoded stimulus intensity and duration. (A) Whole-cell current-clamp recording example of medial olivocochlear (MOC) neuron voltage responses to current injections. Black arrowhead illustrates a double undershoot after-hyperpolarization waveform often observed at action potential threshold-level current injection and is characteristic of MOC neurons. (B) Example of the instantaneous spike rate over time at increasing current injections from the same neuron in (A). Current steps begin at 0.5 nA and increase to 3.0 nA in 0.5 nA steps. (C) Mean spike rate during 500-ms current injections of increasing intensity (N = 33). Averages for current injections up to 900 pA demonstrated a linear input-output curve. A linear function was fit to the mean data, and the y-intercept was forced to 0 pA ($slope = 0.150 \frac{Hz}{pA}$, $r^2 = 0.972$). (D) Same neurons from (C), with current injections up to 4 nA. (E) The ratio of spike-rate adaptation during the last five spikes compared to spikes #1–5 (N = 11). Analysis was not performed on current steps below 200 pA for panels (E) and (F), as no MOC neuron met the minimum requirement of 15 action potentials at those current intensities. (F) The mean spike rate of spikes #5–10 and the mean spike rate of the last five spikes. All error bars are ± SEM (N = 11). Linear functions were fit to the average spike rate of spikes #5–10 ($y = 0.255x - 4.69$, $r^2 = 0.999$) and the average spike rate of the last five spikes ($y = 0.158x - 9.95$, $r^2 = 0.997$).

increasing spike rates, such that the rate nearly doubled when current injections were doubled in intensity, as reflected by the slope of a linear fit to the mean data (slope = 0.150 Hz/pA) (*Figure 2C*). Many MOC neurons continued to respond linearly to current injections up to 2–4 nA (*Figure 2A & D*) before entering depolarization block. Throughout the duration of these 500-ms

depolarizing current injections, action potentials fired with a generally consistent instantaneous rate (*Figure 2B & E*). The ratio of instantaneous spike rate during the last five action potentials (i.e., the steady-state frequency) compared to spikes #5–10 (initial frequency) decreased somewhat with increasing current intensity (*Figure 2E & F*); 0.86 ± 0.01 at 200 pA and 0.69 ± 0.02 at 900 pA (*N* = 11). However, individual cells linearly encoded current intensity with both their initial and their steady-state instantaneous spike frequencies (initial slope = 0.255 Hz/pA; steady-state slope = 0.158 Hz/pA) (*Figure 2F*). These results using current steps suggest that MOC neurons are well suited to delivering steady efferent signals to the cochlea in exact proportion to the intensity of their ongoing synaptic inputs. Therefore, we next explored the properties of synaptic inputs to MOC neurons to determine how this intrinsic firing capacity is utilized under more physiological conditions.

## Light-evoked EPSCs produced by ascending ventral cochlear nucleus inputs are due to fast-gating, inwardly rectifying AMPARs

To activate the excitatory ascending VCN input onto MOC neurons, the VCN was unilaterally infected with 50 nl of AAV-expressing channelrhodopsin (ChR2) fused to the fluorophore Venus (AAV1-CAG-ChR2-Venus-WPRE-SV40) (*Figure 3—figure supplement 1A*). These injections resulted in Venus expression in VCN (*Figure 3A*) and Venus-positive fiber projections to the contralateral VNTB and rostral periolivary regions (RPO) (*Figure 3B*); moreover, Venus-positive boutons were observed in close proximity to MOC neuron dendrites and somata in the VNTB (*Figure 3C*). Loose-patch recordings were conducted in acute brain slices on Venus-positive VCN neurons to determine if potentials mediated by light activation of virally transduced ChR2 would reach action potential threshold at the level of the soma. Venus-positive VCN neurons reliably fired action potentials in response to repetitive 2 ms flashes of blue light (see example, *Figure 3D*). Additionally, reliable antidromic spiking of VCN neurons was observed when light flashes were delivered to axons outside the field of view of the recorded neuron (*N* = 4). The postsynaptic effects of activation of ChR2 were then examined in whole-cell recordings from MOC neurons made in the presence of 5 μM strychnine and 10 μM SR95531 to block inhibitory receptors, and 10 μM MK-801 to block *N*-methyl-D-aspartate (NMDA) receptors. Light-evoked excitatory postsynaptic currents (EPSCs) were observed in MOC neurons both ipsi- and contralateral to the injection site, and the data were combined. EPSCs were abolished with a selective non-competitive AMPA receptor antagonist, GYKI 53655 (50 μM, *N* = 3; not shown), indicating that they were glutamatergic and used AMPA receptors. For individual neurons held at −62.8 mV, 20 light-evoked EPSCs were averaged and their decay phases were best fit with either a single (*N* = 7) or double (*N* = 5) exponential equation. The decay time constant ($\tau$) of double exponential fits was reported as a fast decay component ($\tau_{fast}$) and a slow decay component ($\tau_{slow}$); see *Table 1*. For comparison between double and single exponential fits, $\tau_{fast}$ and $\tau_{slow}$ were converted to a weighted decay time constant.

$$\tau_w = \tau_{fast} * \%A_{fast} + \tau_{slow} * \left(1 - \%A_{fast}\right), \text{ where } \%A_{fast} = \frac{A_{fast}}{A_{slow} + A_{fast}}.$$

$A_{fast}$ and $A_{slow}$ are the absolute amplitudes of each component. There was no significant difference between $\tau$ from single exponential fits and $\tau_w$. Current-voltage (I-V) relations were constructed by plotting the peak amplitude of EPSCs evoked at holding potentials between −82.8 and +57.2 mV (20 mV steps) and exhibited prominent inward rectification (*Figure 3E–F*). The voltage sensitivity of the peak currents, together with the fast decay of the EPSCs, is strongly suggestive of postsynaptic GluA2-lacking $Ca^{2+}$-permeable AMPARs (CP-AMPARs) (*Mosbacher et al., 1994*; *Bowie and Mayer, 1995*; *Donevan and Rogawski, 1995*; *Geiger et al., 1995*).

## Selective activation of T-stellate neurons using an intersectional AAV approach

The results described above indicate that inputs from VCN-originating axons generate EPSCs in MOC neurons, but do not indicate which subtype of VCN neuron is involved. Given the presence of multiple subtypes of VCN excitatory neurons, and the absence of selective Cre lines for these subtypes, a definitive demonstration of the source of input to MOC neurons is challenging. T-stellate (also called planar multipolar) cells of the VCN are excitatory projection neurons that receive auditory nerve inputs (*Oertel et al., 2011*). As a population, they may encode sound intensity and frequency spectrum. T-stellate cells are a major ascending pathway of the auditory system that projects widely to many targets and are the only VCN cell type that projects to the IC.

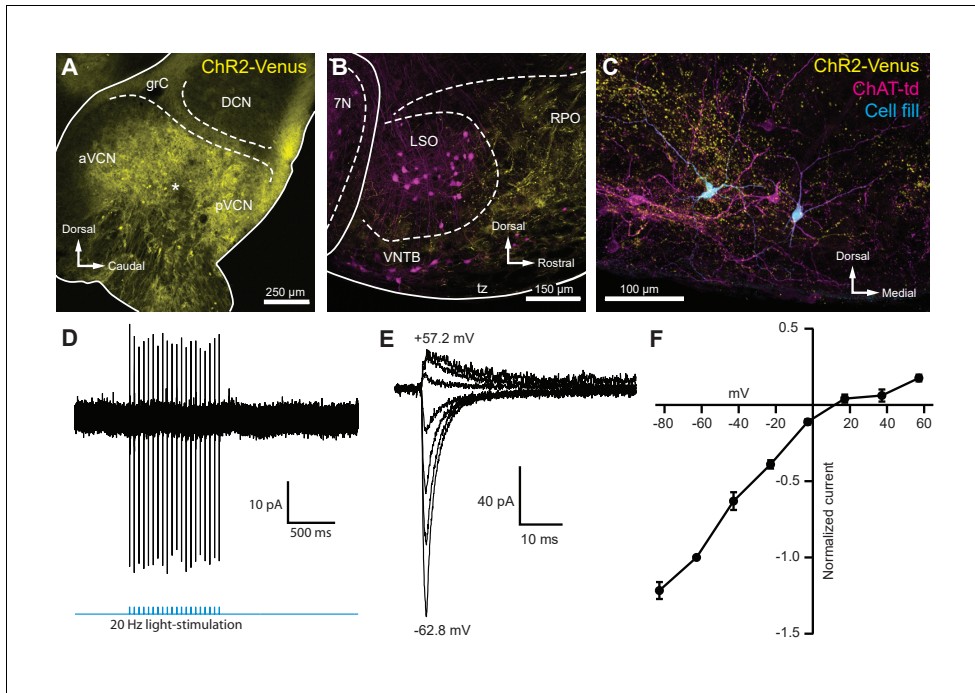

**Figure 3.** Light-evoked EPSCs produced by ascending cochlear nucleus inputs were due to inwardly rectifying AMPARs. (**A**) Sagittal micrograph of a ChR2-Venus-positive ventral cochlear nucleus (* marks the presumed injection site). (**B**) Micrograph from the same mouse as in (**A**), where ChR2-positive fibers are present in the VNTB and RPO near MOC neuron somata. (**C**) Two MOC neurons in the VNTB that were recorded from a coronal brain section and filled with biocytin after post-hoc histochemistry. (**D**) Example of loose-patch cell-attached recording of a ChR2-Venus-positive neuron in the VCN. Neurons positive for ChR2-Venus can reliably fire action potentials in response to light stimuli. (**E**) An example of EPSCs evoked during voltage clamp, with holding potentials ranging from −62.8 mV to +57.2 mV in 20 mV steps. Each sweep was baselined to 0 pA and low-pass Bessel filtered at 3000 Hz. (**F**) I-V relation of normalized cumulative data (*N* = 3–9 per mean). Error bars are ± SEM. Abbreviations: VCN, ventral cochlear nucleus; aVCN, anteroventral cochlear nucleus; pVCN, posteroventral cochlear nucleus; DCN, dorsal cochlear nucleus; grC, granule cell layer; 7N, facial motor nucleus; RPO, rostral periolivary region; MOC, medial olivocochlear; VNTB, ventral nucleus of the trapezoid body.

The online version of this article includes the following figure supplement(s) for figure 3:

**Figure supplement 1.** AAV injection schemes to target ascending or descending inputs to MOC neurons.

Several reports suggest that T-stellate cells serve as an excitatory interneuron in the MOC reflex pathway (*Thompson and Thompson, 1991*; *de Venecia et al., 2005*; *Darrow et al., 2012*), although there is currently no direct evidence for functional connectivity between T-stellate cells and MOC

**Table 1.** Decay time constants for evoked and miniature EPSCs.

| | Double exponential | | | | | | Single exponential |
|---|---|---|---|---|---|---|---|
| | $\tau_{fast}$ (ms) | $\tau_{slow}$ (ms) | $A_{fast}$ (pA) | $A_{slow}$ (pA) | %$A_{fast}$ (*100) | $\tau_w$ (ms) | $\tau$ (ms) |
| VCN input N = 12 | 0.75 ± 0.26 (7) | 4.11 ± 0.86 (7) | −78.63 ± 21.10 (7) | −37.98 ± 13.10 (7) | 70.79 ± 0.25 (7) | 1.59 ± 0.25 (7) | 2.22 ± 0.68 (5) |
| T-stellate input N = 4 | 0.62 (1) | 3.32 (1) | −131.60 (1) | −54.12 (1) | 70.86 (1) | 1.19 (1) | 1.73 ± 0.57 (3) |
| IC input N = 28 | 0.63 ± 0.09 (13) | 3.43 ± 0.30 (13) | −73.09 ± 16.55 (13) | −45.44 ± 9.13 (13) | 60.64 ± 5.00 (13) | 1.72 ± 0.22 (13) | 2.09 ± 0.28 (15) |
| mEPSC N = 3 | 0.17 ± 0.01 (3) | 1.72 ± 0.43 (3) | −47.07 ± 5.36 (3) | −5.63 ± 1.37 (3) | 89.04 ± 3.00 (3) | 0.32 ± 0.02 (3) | N/A |

$\tau_w = \tau_{fast} * \%A_{fast} + \tau_{slow} * (1 - \%A_{fast})$, $\%A_{fast} = \frac{A_{fast}}{A_{fast} + A_{slow}}$. Number of cells per data point denoted as (N).

neurons. We developed a scheme to selectively activate T-stellate cells using an intersectional AAV approach in order to perform virally driven optogenetic studies of ascending MOC circuitry (*Figure 4Ai–ii*). An AAV engineered to infect axons in addition to neurons local to the injection site (AAVrg-pmSyn1-EBFP-Cre) (*Tervo et al., 2016*) was injected into IC of ChAT-Cre/tdTomato mice, causing Cre-dependent tdTomato expression in cells that project to and from IC, including T-stellate cells in the VCN (*Figure 4Ai* and Bii). Prior to AAV infection, no somata were positive for tdTomato in VCN or IC of ChAT-Cre/tdTomato mice (*Figure 4Bi*, *Figure 4—figure supplement 1A*). 1–2 weeks post IC infection with AAVrg-pmSyn1-EBFP-Cre, tdTomato-positive somata were located near the injection site and nuclei that send projections to IC, including contralateral IC (*Figure 4—figure supplement 1B*). A majority of retrogradely labeled VCN somata were located contralateral to the injection site (*Figure 4Bii*), whereas few were seen in ipsilateral VCN (*Figure 4—figure supplement 1C*), reflecting previously described ipsilateral T-stellate cell projections (*Adams, 1979*; *Thompson, 1998*).

In recordings from tdTomato-positive VCN neurons ($N$ = 15) in AAVrg-pmSyn1-EBFP-Cre-infected ChAT-Cre/tdTomato mice, all neurons exhibited responses to current injections that were characteristic of T-stellate cells (see example in *Figure 4C*). Action potentials fired tonically with a sustained rate in response to depolarizing current injections (*Figure 4D*). Hyperpolarizing current injections revealed a rectifying voltage response characteristic of HCN nonselective cation channels (*Figure 4C*). Additionally, membrane resistance ($R_m$ = 147.6 ± 21.5 MΩ) and membrane capacitance ($C_m$ = 32.9 ± 2.6 pF) were typical of T-stellate cells (*Wu and Oertel, 1987*; *Ferragamo et al., 1998*; *Golding et al., 1999*).

A second virus that expressed Cre-dependent ChR2 and enhanced yellow fluorescent protein (EYFP) was then injected into the VCN, enabling ChR2 and EYFP expression only in T-stellate cells that project to contralateral IC (*Figure 4Aii, E–F*). VCN neurons positive for EYFP were also positive for tdTomato (*Figure 4Ei–F*), confirming the selectivity of this intersectional AAV approach. Dual infected VCN neurons projected to known T-stellate cell target nuclei, including contralateral IC, contralateral and ipsilateral VNTB, ipsilateral LSO, and contralateral lateral lemniscus (*Figure 4—figure supplement 1D-I*). During whole-cell voltage-clamp recording, optogenetic activation of T-stellate inputs evoked EPSCs in contralateral MOC neurons ($N$ = 4, *Figure 4G*), confirming that T-stellate neurons excite postsynaptic MOC efferent neurons. Decay kinetics of T-stellate inputs to MOC neurons were not significantly different compared to non-specific VCN inputs (*Table 1*), suggesting similar postsynaptic AMPA receptor compositions. These results definitively show that at least a subset of IC-projecting T-stellate cells provide glutamatergic excitatory inputs to MOC neurons.

## Light-evoked EPSCs produced by descending inferior colliculus inputs are due to fast-gating, inwardly rectifying AMPARs

To activate excitatory descending IC inputs onto MOC neurons, the IC of ChAT-Cre/tdTomato mice were unilaterally infected with 100 nl of AAV1-CAG-ChR2-Venus-WPRE-SV40 (*Petreanu et al., 2009*; *Figure 3—figure supplement 1B*), an anterograde-transported viral construct. 1–2 weeks post-infection, Venus was observed in somata throughout the injected IC (*Figure 5A*). The majority of Venus-positive fibers were visible in the ventral portion of the VNTB and RPO in close apposition to MOC neuron somata and dendrites (*Figure 5B*, *Figure 5—figure supplement 1*). Loose-patch cell-attached recordings of Venus-positive neurons in the IC were conducted to assess ChR2 expression. IC neurons positive for Venus fired action potentials in response to 2 ms flashes of blue light (see example in *Figure 5C*, $N$ = 2), confirming that ChR2 currents could reliably elicit action potentials in response to high-frequency light stimuli. Similar to VCN inputs onto MOC neurons, evoked EPSCs originating from IC inputs were mediated by inwardly rectifying AMPARs (*Figure 5D–E*). This suggests that IC and VCN synapses onto MOC neurons both transmit by means of postsynaptic CP-AMPARs.

## Inward rectification is due to endogenous polyamine block and Ca²⁺-permeable AMPARs

GluR2-lacking CP-AMPARs show rapid decay kinetics and inward rectification due to voltage-dependent block by intracellular polyamines (*Bowie and Mayer, 1995*; *Donevan and Rogawski, 1995*).

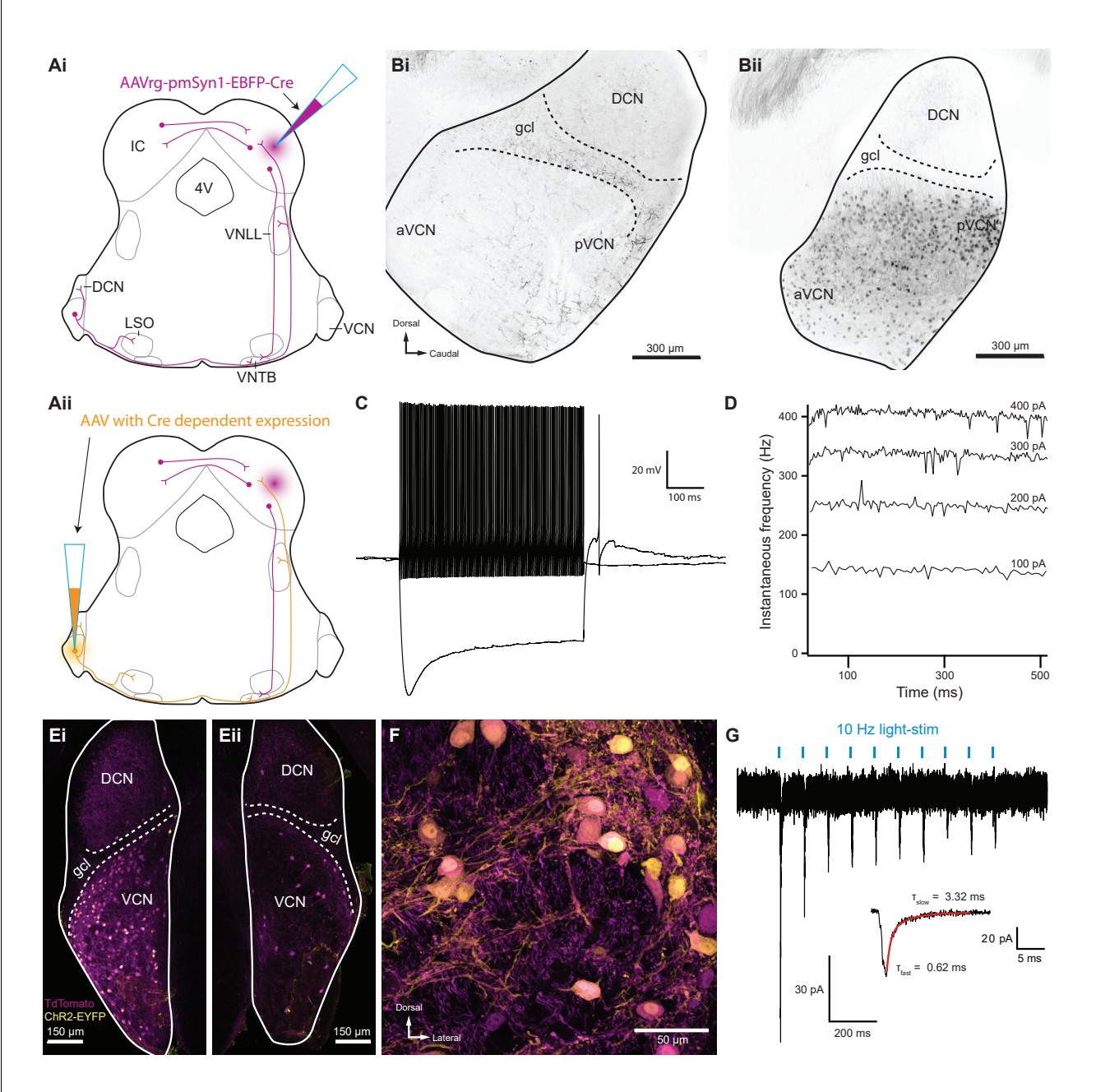

**Figure 4.** Inferior colliculus projecting T-stellate neurons synapsed onto MOC neurons in the ventral nucleus of the trapezoid body. (**Ai**) Schematic depicting an inferior colliculus injection site of Cre recombinase-expressing retrograde AAV (AAVrg-pmSyn1-EBFP-Cre) and putative retrogradely infected neurons and projections (magenta), including T-stellate cells and descending IC projections to olivocochlear efferents. (**Aii**) Continuation of (**Ai**), depicting the ventral cochlear nucleus injection site for a second AAV expressing a Cre-dependent channelrhodopsin. T-stellate neurons (orange) positive for both AAVs project to the VNTB, where MOC neuron somata reside. (**Bi**) Sagittal micrograph of a ChAT-Cre/tdTomato cochlear nucleus. TdTomato-positive fibers were visible throughout the nucleus; however, there was a complete lack of tdTomato-positive somata in VCN. (**Bii**) Sagittal micrograph of a ChAT-Cre/tdTomato cochlear nucleus infected with Cre-expressing retrograde AAV that was injected into the contralateral IC. TdTomato-positive somata were visible throughout the VCN. (**C**) Example of a current-clamp whole-cell recording from an AAVrg-pmSyn1-EBFP-Cre/ tdTomato-positive cell in the VCN. All recordings from tdTomato-positive cells in VCN (*N* = 13) exhibited responses to current injections characteristic of T-stellate cells. Action potentials fired tonically with a sustained rate in response to depolarizing current injections (0.2 nA). Hyperpolarizing current injections (−0.5 nA) revealed a rectifying voltage response. (**D**) Example plot of instantaneous frequency of action potentials throughout the duration of depolarizing stimuli ranging from 100 to 400 pA. The spike frequency is sustained throughout the duration of the stimulus, which is characteristic of T-stellate neurons in the VCN. (**Ei**) Coronal micrograph of a cochlear nucleus contralateral to Cre-expressing retrograde AAV infection of IC. The VCN

*Figure 4 continued on next page*

*Figure 4 continued*

contralateral to IC infection was additionally infected with AAV2-EF1a-DIO-hChR2(E123T/T159C)-p2A-EYFP (UNC Vector Core), which expressed EYFP in the cytosol. (**Eii**) Coronal micrograph of a cochlear nucleus ipsilateral to Cre-expressing retrograde AAV infection of IC from the same mouse as in (**Ei**). Many somata positive for both tdTomato and EYFP are seen in VCN contralateral to the IC infection in (**Ei**), whereas no cells positive for EYFP are seen in VCN ipsilateral to the IC infection in (**Eii**). (**F**) A micrograph of VCN contralateral to Cre-expressing retrograde AAV infection of the IC depicting tdTomato and EYFP expression from the same mouse as in (**Ei**) and (**Eii**). (**G**) Whole-cell voltage-clamp recording from an MOC neuron of a ChAT-Cre/tdTomato mouse contralateral to T-stellate cells expressing ChR2 via the intersectional AAV approach. Light-evoked EPSCs were stimulated with 2 ms pulses of blue light at 10 Hz, and in this example, the EPSCs were depressed with repetitive stimulation. This trace was created by averaging 20 sweeps of the same protocol from the same MOC neuron. Inset illustrates fast and slow decay time constants (τ) of the first averaged EPSC and were fit with a double exponential function (red). Abbreviations: AAV, adeno-associated virus; EYFP, enhanced yellow fluorescent protein; IC, inferior colliculus; 4V, fourth ventricle; VNLL, ventral nucleus of the lateral lemniscus; DCN, dorsal cochlear nucleus; VCN, ventral cochlear nucleus; aVCN, anteroventral cochlear nucleus; pVCN, posteroventral cochlear nucleus; LSO, lateral superior olive; MOC, medial olivocochlear; VNTB, ventral nucleus of the trapezoid body.

The online version of this article includes the following figure supplement(s) for figure 4:

**Figure supplement 1.** Intersectional AAV injection micrographs and T-stellate cell targets.

Demonstration of such block by polyamines could support the interpretation that inputs to MOC neurons are indeed GluR2-lacking. We reasoned that removal of endogenous polyamines by dialysis

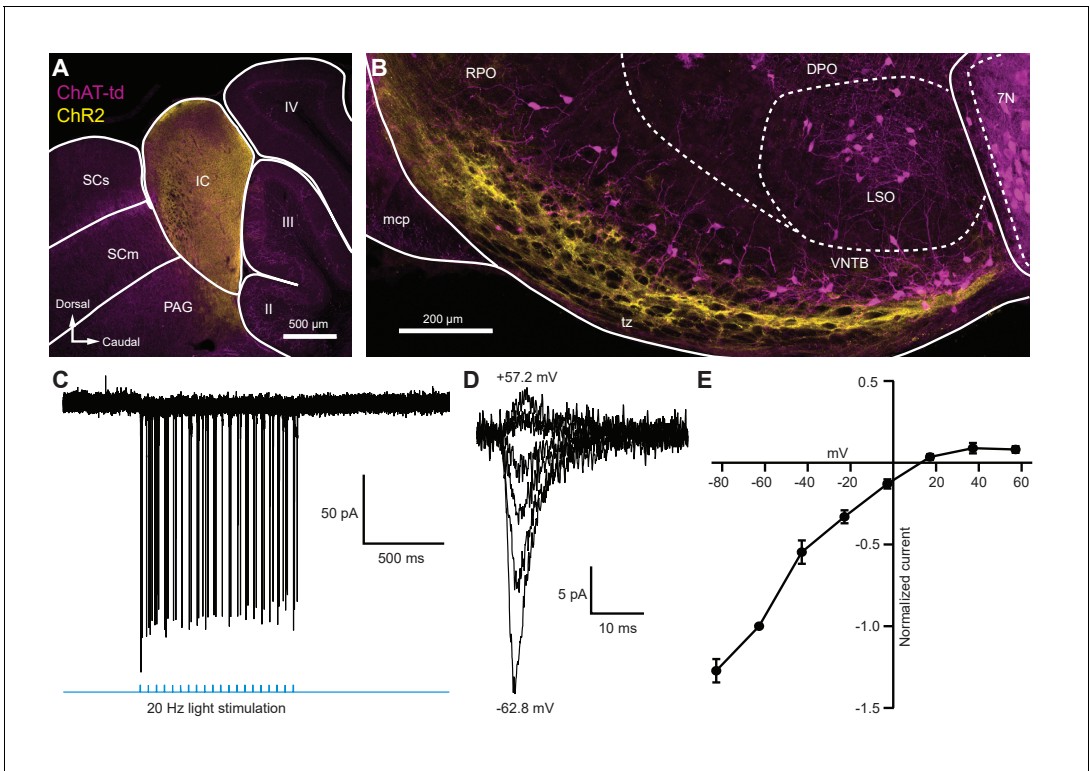

**Figure 5.** Light-evoked EPSCs produced by descending inferior colliculus inputs were due to inwardly rectifying AMPARs. (**A**) Sagittal micrograph of a ChAT-Cre/tdTomato brain section showing an IC injection site positive for ChR2-Venus. (**B**) Sagittal micrograph of the superior olivary complex from the same mouse as in (**A**). The majority of ChR2-positive fibers were visible in the ventral portion of the VNTB/RPO, near MOC neuron somata. (**C**) Loose-patch cell-attached recording of a ChR2-positive neuron in the IC. ChR2-Venus-positive neurons can reliably fire action potentials in response to light stimuli. (**D**) An example of EPSCs evoked during voltage clamp, with holding potentials ranging from −62.8 mV to +57.2 mV in 20 mV steps. (**E**) I-V relation reporting normalized cumulative data (N = 4–7 per mean, N = 2 at +57.2 mV). Error bars are ± SEM. Abbreviations: SCs, sensory superior colliculus; SCm, motor superior colliculus; IC, inferior colliculus; PAG, periaqueductal gray; labeled II-IV, cerebellar lobules; PG, pontine gray; RPO, rostral periolivary region; DPO, dorsal periolivary region; mcp, middle cerebellar peduncle; LSO, lateral superior olive; VNTB, ventral nucleus of the trapezoid body; 7N, facial motor nucleus; MOC, medial olivocochlear.

The online version of this article includes the following figure supplement(s) for figure 5:

**Figure supplement 1.** IC projections to MOC neurons.

would be most effective near the patch pipette and comparatively weak in dendrites where excitatory synapses are likely concentrated. Therefore, we applied glutamate by pressure ejection directly to the soma and tested voltage dependence in recordings in which the patch pipette solution contained or lacked the polyamine spermine (100 µM). In the presence of intracellular spermine, glutamate-evoked currents resulted in an inwardly rectifying I-V relation (*Figure 6A–B*), similar to light-evoked EPSCs from IC or VCN input (*Figure 3E–F* and *5D–E*). When recordings were made with a spermine-free solution, the I-V relation was linear (*Figure 6A–B*). At +37.2 mV and +57.2 mV, 57.9% and 58.9% of the outward current was blocked by spermine, respectively, suggesting that a majority of AMPAR-mediated currents are due to CP-AMPARs.

CP-AMPARs are selectively blocked by IEM 1925 dihydrobromide, which binds to the ion-channel pore in GluA2-lacking receptors more potently than GluA2-containing receptors (*Zaitsev et al., 2011*; *Twomey et al., 2018*). CP-AMPAR block by IEM 1925 is both activity and voltage dependent,

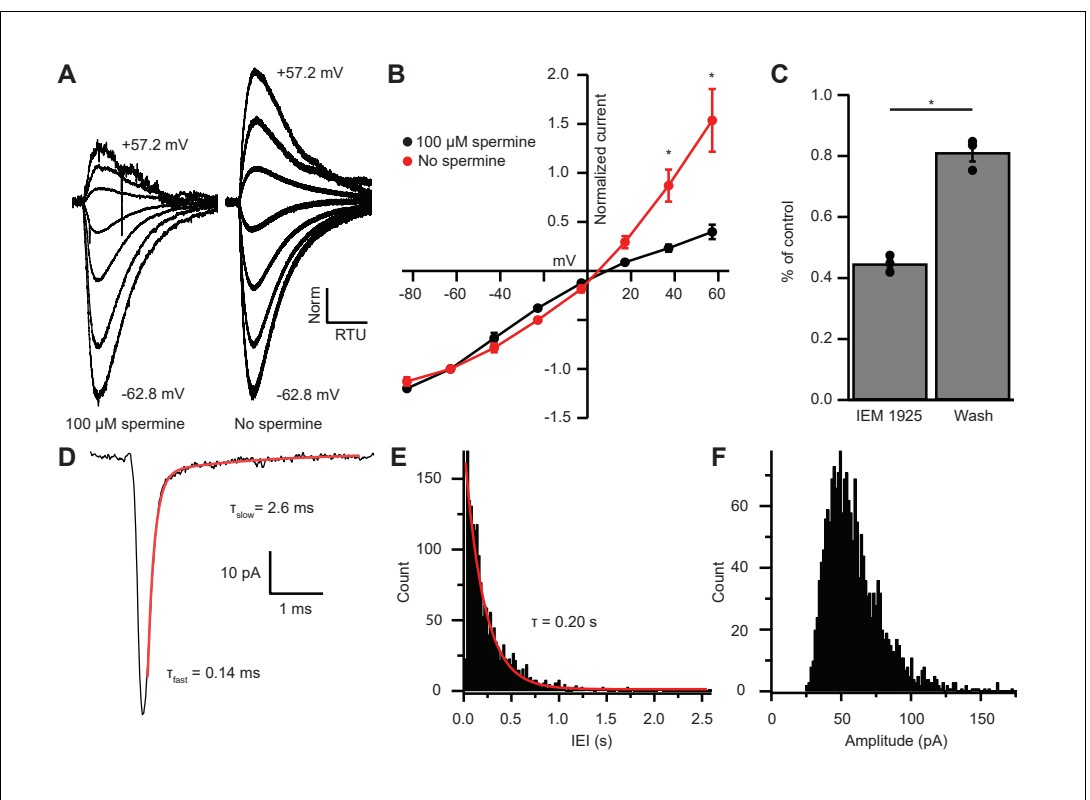

**Figure 6.** EPSC inward rectification was due to endogenous polyamine block and Ca$^{2+}$-permeable AMPARs. (**A**) α-amino-3-hydroxy-5-methyl-4-isoxazolepropionic acid receptor (AMPAR)-mediated currents in medial olivocochlear (MOC) neurons evoked by 1 mM pressure-puffed glutamate near the cell soma. The soma of MOC neurons were dialyzed with an internal pipette solution containing 100 µM or no spermine. In the presence of spermine, glutamate-evoked currents resulted in an inwardly rectifying I-V relation. In the absence of spermine, the rectification was relieved though dialysis, which resulted in a linear I-V relation. Voltage steps ranged from −62.8 mV to +57.2 mV in 20 mV steps; average of 3–10 sweeps per trace. Each sweep was baselined to 0 pA, Bessel filtered at 3000 Hz, and normalized to glutamate-current decays and maximum amplitudes at −62.8 mV. (**B**) An I-V curve showing the average amplitudes (normalized to -62.8 mV) of glutamate-evoked currents in spermine-free (*N* = 3) and 100-µM spermine (*N* = 4) conditions. Error bars are ± SEM. Conditions were significantly different at +37.2 and +57.2 mV (p = 0.019 and 1.2 × 10$^{-6}$, respectively; two-way analysis of variance (ANOVA) with post-hoc Tukey test). (**C**) At -82.8 mV, AMPAR-mediated currents were reduced by 55.29 ± 1.60% with bath application of Ca$^{2+}$-permeable AMPAR antagonist, *N*-(1-phenylcyclohexyl)-1,5-pentanediamine dihydrobromide (IEM 1925; 25 µM). Wash-out of IEM 1925 resulted in inward currents that recovered to 81.20 ± 2.97% of control. (**D**) Average of 582 mEPSCs from one neuron. The fast component (τ$_{fast}$) was responsible for 93.2% of the decay amplitude. Fast decay kinetics are indicative of GluA2-lacking Ca$^{2+}$-permeable AMPARs (CP-AMPARs). (**E**) Inter-event-interval (IEI) distribution of mEPSC activity, 0.02 s bins, 1873 events from three neurons. (**F**) Amplitude distribution of mEPSCs, 1.5 pA bins.

requiring open-state channels and negative potentials. Thus, the amount of block is weakest during spontaneous and evoked synaptic events, and is greatest during the continuous application of agonist. To maximally inhibit CP-AMPAR-mediated currents with IEM 1925, MOC neurons were held at a potential of −82.8 mV in voltage-clamp mode, and 1 mM glutamate was pressure-puffed near MOC neuron somata. After bath application of 25 µM IEM 1925, glutamate-evoked currents were reduced by 55.3 ± 1.6% and returned to 81.2 ± 3.0% of control after wash (*N* = 3, *Figure 6C*). This percentage of block by IEM 1925 was similar to that of spermine block in our dialysis experiments (*Figure 6B*). The blocking effect of IEM 1925 on glutamate-evoked currents pharmacologically confirmed that MOC neurons express GluA2-lacking CP-AMPARs.

## MOC neuron miniature EPSCs are mediated by fast-gating AMPARs

To determine if rapid decay kinetics measured from IC- and VCN-originating EPSCs were synapse specific or a fundamental feature of MOC neuron excitatory synaptic events, we conducted an analysis of miniature EPSCs (mEPSCs). AMPA receptor-mediated currents were pharmacologically isolated and recorded in the presence of 1 µM tetrodotoxin (TTX) to block spontaneous spike-driven events. The decay phase of average miniature events was best fit with a double exponential function, where $\tau_{fast}$ was responsible for 89.0 ± 3.0% of the mEPSC amplitude (see *Table 1* and example in *Figure 6D*). The average $\tau_{fast}$ and $\tau_{slow}$ of mEPSCs were 0.17 ± 0.01 ms and 1.72 ± 0.43 ms, respectively (*N* = 3 neurons, 1873 mEPSCs). The inter-event interval (IEI) between mEPSCs ranged from 6.8 ms to 2.6 s, and each event was counted and sorted into 20 ms bins (*Figure 6E*). The distribution of binned mEPSC IEIs was best described with a single exponential equation ($\tau$ = 0.20 s), reflecting that the miniature events were stochastic in nature (*Fatt and Katz, 1952*). The average and median mEPSC amplitudes were 57.5 ± 0.9 pA and 52.5 pA, respectively, and ranged from 27.2 pA to 146.9 pA (*Figure 6F*). In comparison with light-evoked EPSCs from IC and VCN (*Table 1*), these results confirmed that the majority of mEPSCs were due to fast-gating AMPARs. Together, supportive of data from light-evoked EPSCs and glutamate-puff-evoked currents, these results suggest that fast-gating CP-AMPARs are the major component of excitatory synaptic transmission at MOC neurons.

## Ascending and descending inputs to medial olivocochlear neurons show distinct, opposite forms of short-term plasticity

The AMPARs mediating transmission from VCN and IC were biophysically similar (*Table 1*). However, input-specific repetitive activation of VCN or IC inputs revealed strikingly opposing forms of short-term plasticity (*Figure 7*). During 20-pulse tetanus stimuli (20 or 50 Hz), light-evoked VCN-originating EPSCs depressed whereas IC-originating EPSCs facilitated (*Figure 7A–D*). Plasticity from either input was observed bilaterally in the VNTB and the data were combined. To quantify the change in EPSC amplitude during VCN stimulation, the ratio of the amplitude of the last three EPSCs of the tetanus over the amplitude of the first EPSC was calculated. For IC stimulation, the amplitude of the last three EPSCs of the tetanus was compared to the average amplitude of the first three EPSCs. This 'plasticity index' showed about 70% depression for VCN inputs, with no difference between 20 Hz or 50 Hz activity (0.31 ± 0.02 for 20 Hz, *N* = 8; 0.29 ± 0.04 for 50 Hz, *N* = 7; p = 0.59, Student's *t*-test; *Figure 7C*). By contrast, inputs from IC showed marked enhancement of the plasticity index during the train, although again with no differences between 20 Hz and 50 Hz (1.82 ± 0.17 for 20 Hz, *N* = 8; 1.65 ± 0.26 for 50 Hz *N* = 7; p = 0.59, Student's *t*-test; *Figure 7C*). The degree of plasticity was independent of whether ChR2 was excited near or far from synaptic terminals (*Figure 7—figure supplement 1*).

To analyze recovery from facilitation or depression, a test EPSC was evoked after a 20-pulse tetanus at time intervals increasing from 100 ms to 25.6 s (*Figure 7A and E*). This was repeated 5–20 times for each test pulse with a 30 s or greater gap between sweeps and the results were averaged. We fit recovery data with single exponential functions and found that depression observed by VCN input recovered with a time course ($\tau_{20\ Hz}$ = 3.5 ± 0.7 s, $\tau_{50\ Hz}$ = 3.1 ± 0.4 s) comparable to the recovery from IC input facilitation ($\tau_{20\ Hz}$ = 4.5 ± 1.4 s, $\tau_{50\ Hz}$ = 4.4 ± 1.7 s; *Figure 7E*). While classical short-term facilitation lasts for only hundreds of milliseconds after tetanus stimuli (*Zucker and Regehr, 2002*), IC input facilitation onto MOC neurons lasted for tens of seconds. This longer lasting facilitation resembles synaptic augmentation, which has a longer lifespan (seconds)

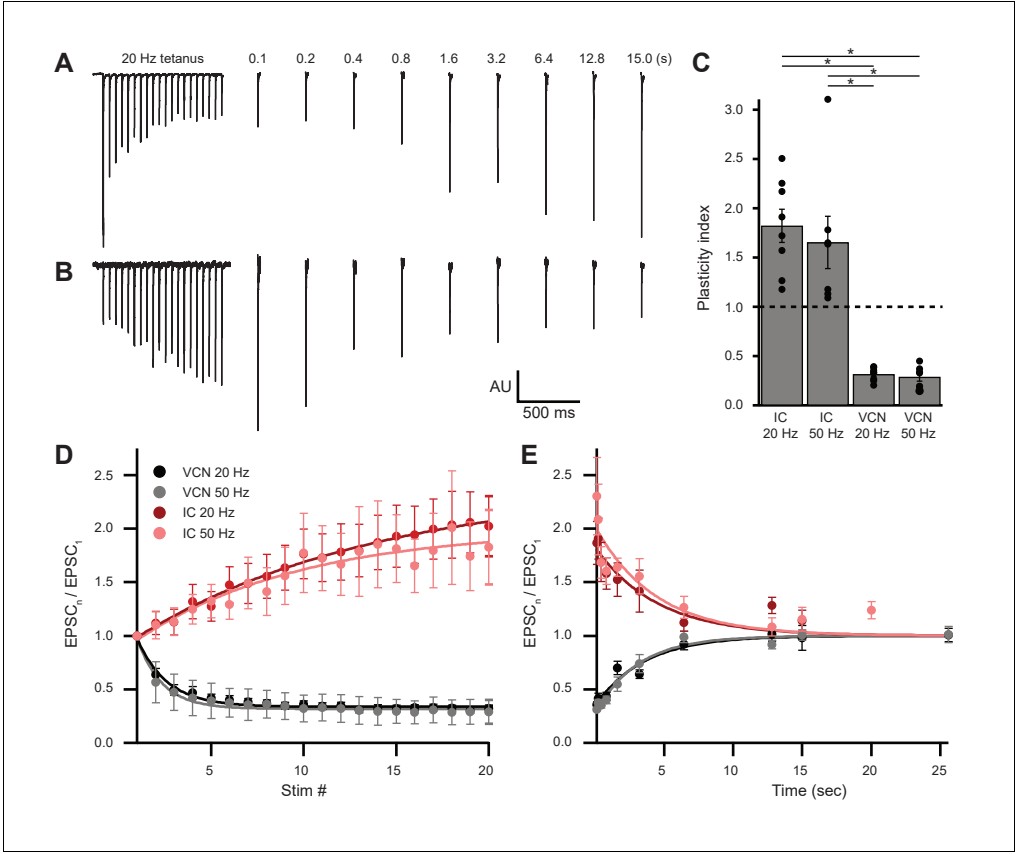

**Figure 7.** Ascending and descending inputs to medial olivocochlear neurons showed distinct short-term plasticity. (**A, B**) Light-evoked EPSCs originating from ventral cochlear nucleus (VCN) (**A**) or inferior colliculus (IC) (**B**) input. During a 20-Hz tetanus stimulus, VCN-originating EPSCs depressed, whereas IC-originating EPSCs facilitated. After each 20-pulse tetanus, a test EPSC was evoked at time intervals increasing from 100 ms to 25.6 s. Each average test EPSC was normalized to the first EPSC of the respective tetanus stimulus. (**C**) 'Plasticity index' to illustrate the degree of facilitation or depression. The index for IC input was the ratio of the amplitude of the last three EPSCs of the tetanus over the amplitude of the first three EPSCs. The index for VCN input was the ratio of the amplitude of the last three EPSCs over the amplitude of the first EPSC of the tetanus. There was no significant difference between 20 Hz and 50 Hz stimulation between inputs of the same origin; however, all IC input was significantly different to all VCN input (p = $4.0 \times 10^{-7}$ at 20 Hz and p = $2.7 \times 10^{-4}$ at 50 Hz; two-way analysis of variance (ANOVA) with post-hoc Tukey test). Error bars are ± SEM. (**D**) Ascending VCN input depresses in amplitude during a tetanus stimulation at both 20 Hz and 50 Hz (20 pulses), while descending IC input facilitates. The average normalized EPSC during a tetanus stimulation is shown for both VCN (N=7, 50 Hz; N=8, 20 Hz) and IC (N=7, 50 Hz; N=8, 20 Hz) inputs. (**E**) Depression observed by ascending VCN input ($\tau_{20Hz}$ = 3.5 ± 0.7 s, $\tau_{50Hz}$ = 3.1 ± 0.4 s) recovered with a similar time course to IC input facilitation ($\tau_{20Hz}$ = 4.5 ± 1.4 s, $\tau_{50Hz}$ = 4.4 ± 1.7 s). The online version of this article includes the following figure supplement(s) for figure 7:

**Figure supplement 1.** Short-term plasticity from VCN and IC inputs onto medial olivocochlear neurons was observed with axonal and terminal level light stimulation.

than classical short-term facilitation (milliseconds) and a recovery time course that is insensitive to the duration or frequency of repetitive activation (*Magleby, 1987*; *Zucker and Regehr, 2002*). Thus, while ascending and descending inputs to MOC neurons employ similar postsynaptic receptors, they differ dramatically in short-term plasticity.

## The onset and dynamic range of MOC neuron output is controlled by integrating facilitating and depressing inputs

We have shown above that the intrinsic properties of MOC neurons permit them to fire over a wide range. Moreover, it has been previously shown that MOC neurons respond dynamically to a wide

variety of binaural sound intensities and frequencies (*Liberman and Brown, 1986*; *Brown, 1989*; *Lilaonitkul and Guinan, 2009*), and thus we expect a large variation in the number of presynaptic fibers driving their output. Given these results, we asked how synaptic inputs from IC and VCN, with their distinct forms of short-term plasticity, utilize the wide firing range of MOC neurons. Our opsin-dependent approach did not allow us to investigate how MOC neurons respond to this presynaptic variation, as one cannot independently control individual fibers in a large population, nor can ChR2 be reliably activated at high, physiological firing rates characteristic of auditory neurons. Thus, we examined how MOC neurons would respond to diverse inputs by injecting synaptic conductance waveforms modeled after physiological data (see 'Materials and methods' and *Figure 8—figure supplement 1*). This approach allowed us to mimic the complex timing of different synaptic inputs that would be difficult to explore with electrical or optical fiber stimulation. A disadvantage is that simulated synaptic conductances are injected in the soma rather than at dendritic synaptic sites, where some local synaptic processing may arise.

To simulate the dynamic firing range of IC and VCN neurons in response to in vivo acoustic stimuli (*Stiebler and Ehret, 1985*; *Rhode and Smith, 1986*; *Smith and Rhode, 1989*; *Kuwada et al., 1997*; *Ono et al., 2017*), we generated low-rate (*Figure 8Ai* and *Figure 8—figure supplement 1Bi* and Ci) and high-rate (*Figure 8Aii* and *Figure 8—figure supplement 1Biii* and Ciii) excitatory postsynaptic conductance (EPSG) waveforms, referred to as ~40 Hz and ~180 Hz, respectively. In mice, the total number of inputs to a single MOC neuron from any region is currently unknown. However, comparing average minimally stimulated light responses to maximal light responses from our in vitro data, we estimated that each MOC neuron in the brain slice receives an average of 4.2 ± 1.0 unilateral inputs from the IC (maximum of 13.8, $N$ = 15) and 11.1 ± 2.4 unilateral inputs from the VCN (maximum of 35.5, $N$ = 15). As these numbers were likely an underestimation due to our experimental preparation (e.g., dependence on virally induced ChR2 expression and damage of inputs during acute brain sectioning), our EPSG waveforms were varied to simulate a broad range of inputs (10, 20, 40, or 80). Additionally, each input's form of presynaptic short-term plasticity could be set to facilitating or depressing, based on our measured parameters. All of our modeled inputs simulated activity of neurons that tonically fire, as we hypothesize that this type of input is most likely to drive sustained responses in MOC neurons. For example, the majority of T-stellate cells in the VCN fire tonically at a constant rate in response to tones (termed 'sustained choppers') (*Oertel et al., 2011*). However, it should be noted that some may also exhibit either a rapid ('transient choppers') or a slowly adapting firing rate (*Blackburn and Sachs, 1989*), including in mouse (*Roos and May, 2012*), suggesting a range of response types not readily apparent in brain slice recording. The cellular identity and intrinsic properties of IC neurons that project to MOC neurons are currently unknown and may exhibit tonic or adapting firing patterns (*Peruzzi et al., 2000*). However, because their nerve terminals in VNTB undergo facilitation to repetitive presynaptic firing, it seems likely that these neurons may exhibit tonic firing, and this was assumed for our simulated facilitating inputs.

In response to a small number of facilitating inputs (labeled 'Fac' in the following figures) firing at ~40 Hz, few action potentials were evoked in MOC neurons (1.8 ± 0.8 for 10 inputs and 5.3 ± 1.5 for 20 inputs, $N$ = 6) (*Figure 8Bi*, D, and E), and the first peak of postsynaptic firing generally occurred hundreds of milliseconds after stimulus onset (271.1 ± 74.4 ms for 10 inputs and 183.8 ± 59.0 ms for 20 inputs, $N$ = 6) (*Figure 8Bi*, D, and F). With only 10 facilitating inputs at ~40 Hz, two out of six MOC neurons failed to reach action potential threshold (e.g., first row of *Figure 9C*). MOC neurons responded to increasing numbers of facilitating inputs, with a linearly increasing number of spikes during each stimulus for both ~40 Hz and ~180 Hz paradigms (*Figure 8E*). Facilitating EPSGs at ~180 Hz generally elicited more action potentials with an earlier onset than ~40 Hz EPSGs with the same number of inputs (*Figure 8Bii,E, and F*). Additionally, the slope (increase in number of spikes for a given increase in number of inputs) of linear fits to the data in *Figure 8E* also significantly increased with presynaptic firing rate; this slope will be referred to as firing sensitivity (FS). FS was 0.31 ± 0.06 for 40 Hz and 1.0 ± 0.1 for 183 Hz (p = $6.7 \times 10^{-6}$; paired samples Student's $t$-test). In a small number of experiments ($N$ = 3), an EPSG waveform would drive an MOC neuron into depolarization block toward the end of each trial, likely due to voltage-gated sodium channel inactivation (e.g., *Figures 8D*, 80 facilitating inputs at ~180 Hz). When this occurred, we measured the average instantaneous frequency (spikes per second) of all action potentials before the onset of depolarization block whose amplitude surpassed a −20 mV threshold and divided this number by half to extrapolate the number of spikes per 500 ms (*Figures 8E* and *9D*).

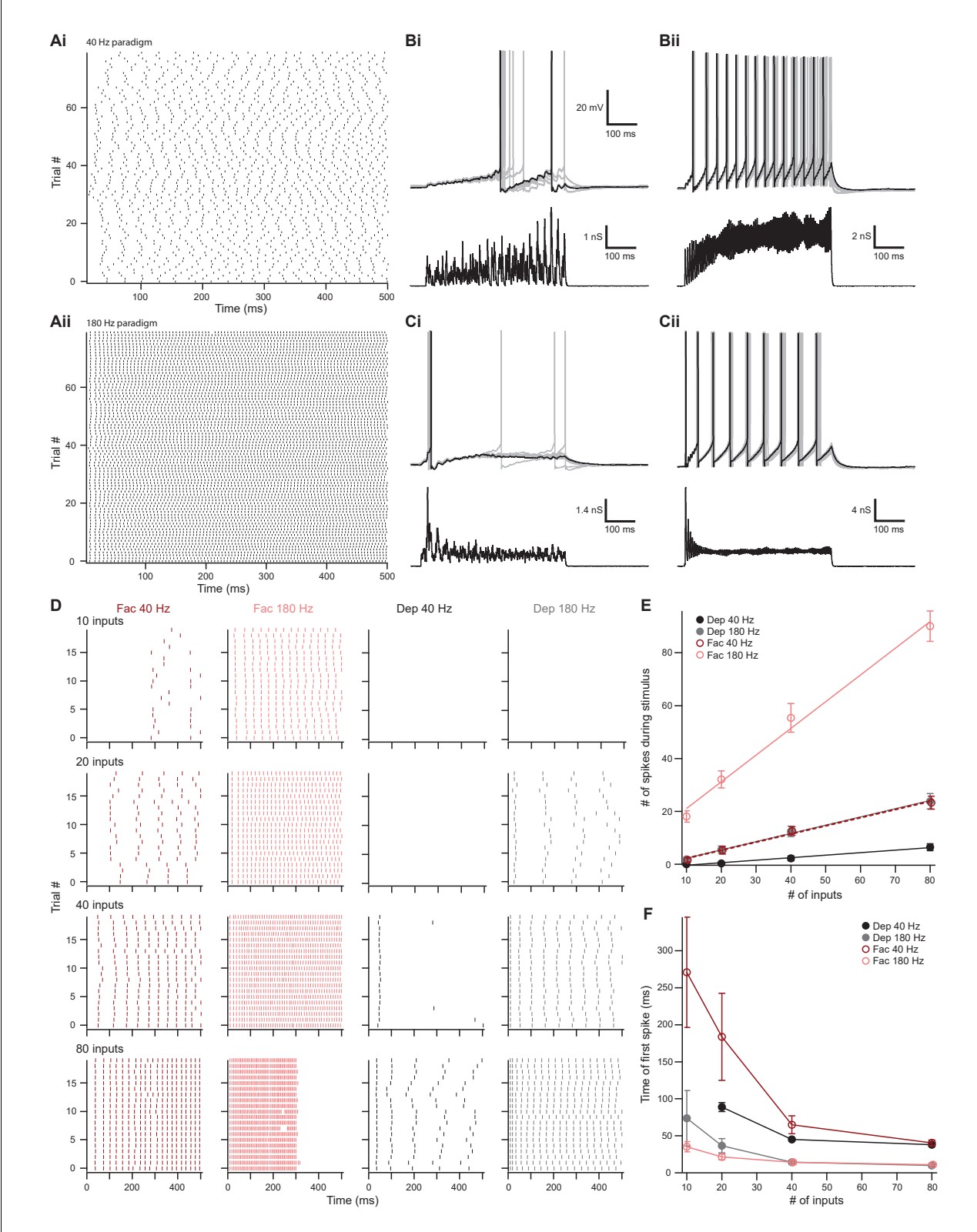

**Figure 8.** The number of presynaptic inputs and type of short-term plasticity control the dynamic range and onset timing of MOC neuron output. (**A**) Raster plots of presynaptic excitatory postsynaptic conductance (EPSG) onset timing. The ~40 Hz paradigm (**Ai**) had an average rate of 41.1 ± 0.5 Hz for all 80 trials. The ~180 Hz paradigm (**Aii**) had an average rate of 176 ± 1 Hz for all 80 trials. Each trial was considered a presynaptic input in our model. (**B**) Ten example traces of membrane voltage responses to injected conductance waveforms simulating 10 inputs at ~40 Hz (**Bi**) or ~180 Hz (**Bii**) that

*Figure 8 continued on next page*

*Figure 8 continued*

underwent short-term facilitation. Scale bar is the same for all voltage responses in (B) and (C). (C) Ten example traces of membrane voltage responses to injected conductance waveforms simulating 40 inputs at ~40 Hz (Ci) or ~180 Hz (Cii) that underwent short-term depression. (D) Example raster plots of postsynaptic medial olivocochlear (MOC) neuron action potential timing in response to injected conductance waveforms. Rows of raster plots correspond to the number of simulated inputs, and columns correspond to the type of simulated presynaptic short-term plasticity and firing rate. Blank raster plots represent an absence of firing. One example (80 presynaptic inputs at ~180 Hz with short-term facilitation) underwent depolarization block after ~300 ms. All examples are from the same MOC neuron. (E) Average total number of action potentials evoked in MOC neurons (N = 6) during each conductance waveform paradigm. Error bars are ± SEM. (F) Average timing of the peak of the first action potential evoked in MOC neurons (N = 6) during each conductance waveform paradigm. Error bars are ± SEM.

The online version of this article includes the following figure supplement(s) for figure 8:

**Figure supplement 1.** Synaptic conductance waveforms were modeled after physiological data.

Similar to facilitating EPSGs, the majority of depressing EPSG waveforms (labeled 'Dep' in the figures) elicited action potentials that fired in a sustained manner (*Figure 8Cii–E*), and the FS in response to EPSG waveforms significantly increased with presynaptic firing rate (*Figure 8E*). FS was 0.095 ± 0.014 for 40 Hz and 0.31 ± 0.08 for 180 Hz (p = $1.6 \times 10^{-4}$, paired samples Student's *t*-test). Some MOC neurons failed to reach action potential threshold in response to depressing waveforms at ~40 Hz (6/6 failures with 10 inputs and 3/6 with 20 inputs) and ~180 Hz (3/6 failures with 10 inputs) (e.g., last two rows of *Figure 8D*). When action potentials were elicited and the number of simulated inputs were equivalent, depressing waveforms at ~180 Hz always drove MOC neurons to threshold earlier than those at ~40 Hz (*Figure 8F*). At ~40 Hz, with 20–40 simulated inputs, depressing waveforms often elicited an onset response (*Figure 8Ci*) that occurred earlier than facilitating waveforms at the same rate (*Figure 8D and F*). When the presynaptic firing rate was increased to ~180 Hz, facilitating waveforms with 10–20 simulated inputs generally elicited an onset response sooner than with depressing inputs. As our previously described experiments demonstrated that inputs from IC facilitated and those from VCN depressed (*Figure 7*), our simulated inputs suggest that, individually, VCN inputs best drive slow rates of sustained activity in MOC neurons and IC inputs best drive high rates of activity. Combinations of these inputs are needed to access the full dynamic range of MOC neuron firing, as our simulated VCN-like (depressing) EPSGs could only drive the firing rate (47.8 ± 5.9 Hz average maximum, N = 6) to about half of the maximum rates measured in vivo (*Liberman, 1988*; *Brown, 1989*), while IC-like (facilitating) EPSGs could drive MOC neurons to fire at maximal rates (180 ± 12 Hz average maximum, N = 6) more comparable to our in vitro experiments (*Figure 2*).

## Descending input to MOC neurons can enhance or override ascending reflex input

The output of MOC neurons in vivo depends on the integration of multiple input subtypes, where weaker ascending inputs may be optimized or overridden by more powerful descending inputs. To investigate how MOC neurons would respond to this type of integration, we injected EPSG waveforms simulating combinations of ascending (depressing) and descending (facilitating) inputs. In order to avoid artificially introducing synchrony between the modeled VCN and IC inputs, we introduced a third average presynaptic firing rate, ~110 Hz (*Figure 9A* and *Figure 8—figure supplement 1Bii* and Cii). Using the ~180 Hz paradigm, 20 depressing inputs elicited a low number of action potentials in MOC neurons without any failures (5.5 ± 1.4 spikes on average, N = 6) (dashed gray line, *Figure 9D*), with the first action potential occurring at 36.5 ± 6.6 ms after the stimuli onset (dashed gray line, *Figure 9E*). To experimentally test how IC-like inputs altered this VCN-like response, we concurrently introduced 10–80 facilitating inputs at ~40 Hz or ~110 Hz (*Figure 9*). As expected, the number of action potentials evoked by facilitating or depressing input was increased when the both types were combined (*Figure 9B–D*). However, the magnitude of this effect was dependent on the strength of the facilitating input, as spikes evoked by weaker facilitating inputs (1.6 ± 0.8 spikes on average for 10 inputs at ~40 Hz and 8.6 ± 2.6 spikes at ~110 Hz) were significantly enhanced when combined with the depressing paradigm (9.2 ± 2.3 spikes at ~40 Hz, 16.1 ± 3.1 spikes at ~110 Hz) (p = 0.022 at ~40 Hz and p = 0.0034 at ~110 Hz; paired samples Student's *t*-test), whereas stronger facilitating inputs were not enhanced (for example, 67.3 ± 8.0 spikes on average for 80 inputs at ~110 Hz vs 68.7 ± 0.4 when combined) (*Figure 9D*). Thus, our modeled

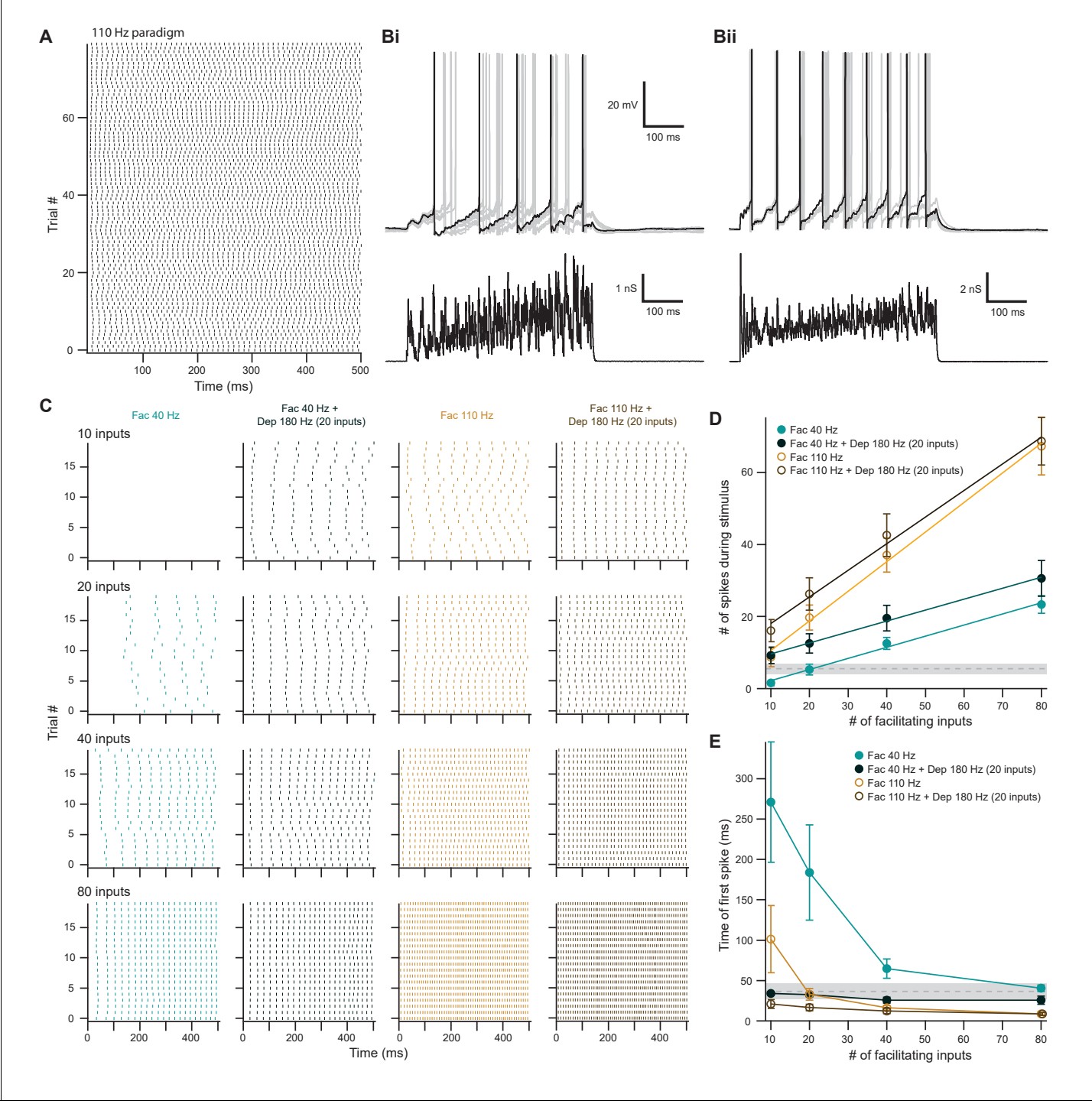

**Figure 9.** Facilitating inputs to MOC neurons can override or be enhanced by depressing inputs, depending on their number and rate. (**A**) Raster plot of presynaptic excitatory postsynaptic conductance (EPSG) onset timing for the ~110 Hz paradigm, which had an average rate of 111 ± 1 Hz for all 80 trials. Each trial was considered a presynaptic input in our model. (**B**) Ten example traces of membrane voltage responses to injected conductance waveforms simulating 20 facilitating inputs at ~40 Hz without (**Bi**) or with (**Bii**) the addition of 20 depressing inputs at ~180 Hz. Scale bar is the same for both voltage responses. (**C**) Example raster plots of postsynaptic medial olivocochlear (MOC) neuron action potential timing in response to injected conductance waveforms, without (even columns) or with (odd columns) the addition of 20 depressing inputs at ~180 Hz. Rows of raster plots correspond to the number of simulated inputs, and columns correspond to the type of simulated presynaptic short-term plasticity and firing rate. Blank raster plots represent an absence of firing. All examples are from the same MOC neuron. (**D**) Average total number of action potentials evoked in MOC neurons (*N* = 5) during each conductance waveform paradigm. Error bars are ± SEM. Gray dashed line represents 20 depressing inputs at ~180 Hz, and the shaded

*Figure 9 continued on next page*

*Figure 9 continued*

area represents ± SEM in (D) and (E). (E) Average timing of the peak of the first action potential evoked in MOC neurons (*N* = 5) during each conductance waveform paradigm. Error bars are ± SEM.

facilitating IC inputs effectively drive MOC firing, and the input from depressing VCN synapses only enhanced firing when the IC input was relatively weak.

Combining facilitating and depressing inputs is expected to impact the timing of postsynaptic action potentials, and so we also examined the onset time of firing. At ~40 Hz, the onset of the first action potential evoked by facilitating inputs occurred earlier when combined with the depressing paradigm, but the strength of this effect decreased with increasing number of facilitating inputs (*Figure 9E*). The same was true for ~110 Hz facilitating inputs when simulating only 10 or 20 inputs. However, there was little difference as the number of inputs increased. The FS of facilitating inputs (0.32 ± 0.06 at ~40 Hz and 0.82 ± 0.18 at ~110 Hz) was not significantly altered with the addition of a depressing input (0.30 ± 0.09 at ~40 Hz and 0.74 ± 0.12 at ~110 Hz, *N* = 5) (p = 0.38 at ~40 Hz and p = 0.088 at ~110 Hz; paired samples Student's *t*-test) (*Figure 9D*), demonstrating that MOC neurons generally summate concurrent inputs and confirming that they linearly respond to signals in proportion to the intensity of their input (*Figure 2*). Overall, these results suggest that relatively weak descending inputs to MOC neurons are enhanced when combined with the ascending input, while relatively strong descending inputs override the ascending input—evoking an equivalent amount of spikes with similar onset, whether or not the ascending input is active. The data, therefore, confirm the potency of the descending control of the MOC system, as compared to the reflex pathway.

## Discussion

In the present study, we contrasted excitatory inputs onto MOC neurons through two distinct sources, an ascending, reflex pathway and a descending pathway from the midbrain. Few studies have explored the properties of MOC neurons, in part due to the difficulty of identifying the neurons in mature, heavily myelinated tissue. In order to overcome previous limitations and visualize MOC neurons in acute brain sections from older mice, we utilized a ChAT-Cre mouse line, which genetically marks cholinergic neurons in the SOC. This line was recently characterized by *Torres Cadenas et al., 2020*, where it was shown to label cholinergic MOC efferent neurons. Using CTB-mediated retrograde tract tracing originating from the inner-ear, we were able to confirm and expand on their results. Agreeing with classic anatomical tract-tracer studies of MOC neurons (for review, see *Warr, 1992*), we demonstrated that approximately two-third of ChAT-Cre/tdTomato-positive VNTB neurons project to contralateral cochlea, whereas one-third project to ipsilateral cochlea. Additionally, retrogradely labeled VNTB neurons were always positive for tdTomato, confirming that the ChAT-Cre mouse line expressed Cre recombinase in most, if not all, MOC efferent neurons in the VNTB.

### Firing rates of MOC neurons

Our examination of intrinsic properties of MOC neurons revealed a remarkable capacity to encode the intensity of current steps with a linear increase in postsynaptic firing over a wide range. Further, we showed that this linearity is recapitulated in the responses to synaptic activity, as modeled through conductance clamp. Previous work by *Fujino et al., 1997* reported MOC and LOC neuron intrinsic membrane properties of neonatal rats (P3-9). However, due to the difficulty of visualizing brainstem neurons in older animals with tract tracers, they were not able to record from MOC neurons after the onset of hearing (P12-14). Additional studies on MOC neurons that used whole-cell recording and tract tracing were also limited to younger animals near or prior to onset of hearing (*Wang and Robertson, 1997*; *Mulders and Robertson, 2001*). Consistent with properties described in prehearing rats (*Fujino et al., 1997*), we reported that a majority of matured MOC neurons did not spontaneously fire and that their spike frequency linearly increased with the intensity of injected current pulses. This conclusion is supported by in vivo recordings at the level of the auditory nerve, where MOC efferents exhibit little-to-no spontaneous firing and respond linearly to increasing sound intensity (*Cody and Johnstone, 1982*; *Robertson, 1984*; *Robertson and Gummer, 1985*). A recent

study (*Torres Cadenas et al., 2020*) reported that MOC neurons from P12-23 mice exhibited spontaneous firing, which may be due to developmental changes specific to mice (we recorded from P30-48) or due to differences in acute brain slice preparation. Developmental transcriptomics of auditory efferent neurons could reveal the basis for these changes.

Intriguingly, in vivo recordings rarely report sound-driven firing rates in MOC efferents above 100 Hz (*Liberman, 1988*; *Brown, 1989*), yet we report that many MOC neurons can fire action potentials at rates greater than 250 Hz in response to somatic current injections. The high firing rates achieved in vitro may better reflect MOC neuron capabilities, as in vivo experiments are often performed with anesthetics that produce extensive systemic changes in neurotransmission (*Robertson and Gummer, 1985*; *Brown, 1989*; *Guitton et al., 2004*; *Chambers et al., 2012*; *Aedo et al., 2015*). An alternative interpretation is that the wide firing range intrinsic to MOC neurons ensures that over the narrower range used in vivo, the linearity of input-output relations remains preserved. MOC neurons are known to receive modulatory inputs from adrenergic, serotonergic, and peptidergic sources (*Thompson and Thompson, 1995*; *Woods and Azeredo, 1999*; *Mulders and Robertson, 2000*; *Mulders and Robertson, 2001*; *Oliver et al., 2000*; *Horváth et al., 2003*), and can be excited by a handful of neuromodulators (*Wang and Robertson, 1997*; *Wang and Robertson, 1998*). This suggests that sound-driven firing rates in MOC neurons observed in vivo may be contextually enhanced by activation of neuromodulatory inputs.

MOC neurons also receive inhibitory inputs that are likely activated by sound (*Torres Cadenas et al., 2020*), but their impact on the in vivo output of this system remains to be explored. It may be that sound-evoked inhibition of MOC neurons could prevent saturation of firing and thereby broaden the input-output relationships we have described. Alternatively, if some of the descending excitation is not acoustically driven but relayed through the IC, then it may be independent of the type of inhibition described in *Torres Cadenas et al., 2020*.

## Excitatory MOC neuron inputs utilize fast-gating CP-AMPARs

The ability of the MOC system to dampen cochlear sensitivity likely depends on the convergence of excitatory synaptic inputs from ascending and descending brain regions. Tract-tracer and lesion studies have determined that ascending projections originate from the posteroventral cochlear nucleus (*Thompson and Thompson, 1991*; *de Venecia et al., 2005*; *Darrow et al., 2012*; *Brown et al., 2013*). These ascending projections are involved in the reflex MOC pathway and are likely mediated by T-stellate neurons. However, bushy cells may also play a role, as in cats they send axon collaterals, which terminate in VNTB, the primary location of MOC neuron somata (*Smith et al., 1991*). Descending projections to MOC neurons originate from auditory and non-auditory regions, including brainstem, IC, thalamus, and cortex (*Thompson and Thompson, 1993*; *Vetter et al., 1993*; *Mulders and Robertson, 2002*). The IC is a major source of dense, tonotopically arranged, glutamatergic projections to ipsilateral VNTB (*Thompson and Thompson, 1993*; *Saint Marie, 1996*; *Suthakar and Ryugo, 2017*), where the majority of IC projections terminate (*Terreros and Delano, 2015*; *Cant and Oliver, 2018*), and its targets include MOC neurons.

In the present study, we elucidated pre- and postsynaptic properties of excitatory VCN and IC inputs onto MOC neurons by using nucleus- and cell-specific virally driven optogenetic excitation. We demonstrated that MOC neurons receive excitatory inputs from VCN and IC, both of which transmit using fast-gating CP-AMPARs. Together with somatic puff application of glutamate and mEPSC analysis, our investigation revealed that inwardly rectifying, fast-gating CP-AMPARs are a fundamental postsynaptic feature of excitatory synaptic transmission at MOC neurons. However, it is unknown if calcium influx through these receptors can induce some form of synaptic plasticity (*Cull-Candy and Farrant, 2021*) or if the receptor kinetics play an important role in postsynaptic transmission.

The utilization of GluR2-lacking AMPARs, with ultra-fast mEPSC decays (less than 200 µs), is reminiscent of auditory nerve synapses in the VCN, including those onto T-stellate cells (*Gardner et al., 1999*; *Gardner et al., 2001*). Higher regions of the auditory pathway typically lack this feature of the synapse, even in the adjacent medial nucleus of the trapezoid body, whose mEPSCs are slower than in MOC neurons and likely contain GluR2 (*Koike-Tani et al., 2005*; *Lujan et al., 2019*). We do not know if the mEPSCs originated from synapses made by VCN or IC neurons or both, but the uniformity of mEPSC properties suggests that even descending fibers from IC can trigger activation of such fast-gating receptors. While the presence of fast kinetic receptors is considered to be an

adaptation to preserve microsecond precision of sensory timing (*Gardner et al., 1999*), it seems unlikely that such a precise timing is needed in the efferent system. Further studies are needed to examine how receptor channel kinetics impact the integrative functions of the MOC neuron.

## T-stellate neurons are an MOC reflex interneuron

Neurons identified as T-stellate cells are believed to terminate in VCN, DCN, olivary nuclei, lemniscal nuclei, and IC (*Warr, 1995*; *Oertel et al., 2011*), but it is not clear if axons of the same neuron can have such diverse projections. Using an intersectional AAV approach, we directly demonstrated that T-stellate neurons drive activity in MOC neurons, consistent with suggestions from previous anatomical and lesion studies (*Thompson and Thompson, 1991*; *de Venecia et al., 2005*; *Darrow et al., 2012*). T-stellate projections and terminals in many known target nuclei were consistently observed in brain sections prepared for microscopy (*Figure 4—figure supplement 1*). Nevertheless, eliciting a postsynaptic current was qualitatively difficult when compared to non-specific virally mediated ChR2 expression in the VCN. This was possibly due to sparse ChR2 expression among T-stellate neurons resulting from the requirement of coincident infection by two different viruses in the same neuron; alternatively, ChR2 expression may have been too low to consistently reach action potential threshold using the intersectional AAV scheme. There also may be sub-populations of T-stellate neurons, which project to MOC neurons and do not project to the IC; non-IC-projecting T-stellate neurons would not express ChR2 using this intersectional virus approach. Nevertheless, this approach highlights the enormous range of targets of these neurons, as at least a subset of IC-projecting T-stellate neurons also directly synapsed onto MOC neurons. Genetic manipulation of only T-stellate neurons with this dual-AAV approach will be useful in future studies to help elucidate the functional significance of T-stellate projections in other auditory circuits.

## Effect of short-term synaptic plasticity on MOC neuron output

Neurons throughout the brain receive mixtures of synaptic inputs that vary not only in their origin or information content, but also in their short-term plasticity. A prominent example is that of cerebellar Purkinje neurons, whose parallel fiber inputs facilitate while climbing fiber inputs depress (*Sakurai, 1987*; *Hansel and Linden, 2000*). The physiological functions served by this diversity likely vary with brain region. In MOC neurons, we found that synaptic responses having properties of the ascending or descending inputs alone were not capable of encoding firing over a wide range and with short latency. However, by combining these different types of inputs and varying input number and firing rate, sustained MOC output could vary over twentyfold. We suggest that this central synaptic mechanism could aid in grading the level of efferent dampening of cochlear function according to sound level.

Inputs from IC strengthened considerably for tens of seconds with repetitive presynaptic stimulation, resulting in a facilitation that resembles the augmentation seen at neuromuscular junctions (*Magleby and Zengel, 1976*; *Figure 7*), whereas VCN and T-stellate inputs (*Figure 4G*) decreased in synaptic strength, resulting in acute short-term depression. Our optogenetic stimulation was limited in frequency, and further studies will need to employ faster opsins in order to explore plasticity at still higher frequencies of stimulation. Both forms of plasticity that we observed recovered over a similar time course, suggesting that conditioning of these synapses could have lasting effects and bias efferent signaling toward top-down control. The depression of VCN inputs to MOC neurons is not likely due to desensitization of ChR2, since trains of light pulses triggered reliable spikes in VCN neurons. Moreover, injections into IC were made with the same virus, and those inputs never exhibited depression. Thus, distinct forms of presynaptic plasticity are likely exhibited by IC and VCN inputs to the same cell type. Depression of VCN inputs is surprising, given that these inputs mediate a reflex pathway and one might therefore expect reliability within such a circuit. Moreover, as with MOC neurons, the majority of T-stellate neurons fire action potentials in a relatively sustained manner in response to sound stimuli. In in vivo recordings at the level of the auditory nerve, MOC neurons respond to sound input with latencies as short as 5 ms (*Robertson and Gummer, 1985*; *Liberman and Brown, 1986*), and T-stellate cells are well suited to provide the rapid onset portion of this response, as demonstrated in our simulation of this input (*Figure 8*). However, our results suggest that for sustained activity of MOC efferents, non-VCN inputs, such as from the IC, may be a necessary component of efferent control of cochlear function. Indeed, a recent auditory system

computational model suggested that descending IC inputs to the MOC system are necessary for persistent enhancement of signal in noise and that the MOC system functions across a broad range of intensity (*Farhadi et al., 2021*). These features of the model are now affirmed by our observations of potent inputs from IC, dependent on synaptic augmentation, and the intrinsic properties of MOC neurons that support a remarkably wide dynamic range. The diversity of cell types in the IC is a topic of current interest, and it will be important to determine which of these mediates descending control of the brainstem, including the efferent system, and which ascending or descending pathways activate these neurons. Indeed, it may be that the stable, excitatory control of efferent neurons by the descending input raises the possibility that regulation of cochlear sensitivity may be under rapid control associated with attention (*Delano et al., 2007*; *Wittekindt et al., 2014*), preceding sounds (*Otsuka et al., 2018*), or other changes in brain state.

# Materials and methods

## Key resources table

| Reagent type (species) or resource | Designation | Source or reference | Identifiers | Additional information |
|---|---|---|---|---|
| strain, strain background (*M. musculus*) | ChAT-IRES-Cre | Jackson Laboratory PMID:21284986 | RRID:IMSR_JAX:006410 | |
| strain, strain background (*M. musculus*) | Ai9(RCL-tdT) | Jackson Laboratory PMID:22446880 | RRID:IMSR_JAX:007909 | |
| strain, strain background (*M. musculus*) | C57BL/6J | Jackson Laboratory | RRID:IMSR_JAX:000664 | |
| Chemical compound, drug | Strychnine hydrochloride | Sigma | Cat# S8753 | |
| Chemical compound, drug | SR-95531 hydrobromide | Tocris | Cat# 1262 | |
| Chemical compound, drug | NBQX disodium salt | Tocris | Cat# 1044 | |
| Chemical compound, drug | (+)-MK-801 hydrogen maleate | Sigma Aldrich | Cat# M107 | |
| Chemical compound, drug | Tetrodotoxin (TTX) | Sigma Aldrich | CAT# 554412 | |
| Chemical compound, drug | Biocytin | ThermoFisher Scientific | Cat# B1592 | |
| Chemical compound, drug | Cholera toxin subunit B | List Labs | Cat# 104 | |
| Antibody | anti-ChAT (Goat polyclonal) | Millipore | Cat# AB144P RRID:AB_2079751 | IHC (1:500) |
| Antibody | anti-GFP (Chicken polyclonal) | Aves Labs | Cat# GFP-1020 RRID:AB_10000240 | IHC (1:1000) |
| Antibody | anti-Cholera Toxin B Subunit (Goat polyclonal) | List Labs | Cat# 703 RRID:AB_10013220 | IHC (1:1000) |
| Antibody | anti-DsRed (Rabbit polyclonal) | Clontech | Cat# 632496 RRID:AB_10013483 | IHC (1:500) |

*Continued on next page*

*Continued*

| Reagent type (species) or resource | Designation | Source or reference | Identifiers | Additional information |
|---|---|---|---|---|
| Antibody | anti-chicken Alexa Fluor 488 (Donkey polyclonal) | Jackson Immuno Research Labs | Cat# 703-545-155RRID:AB_2340375 | IHC(1:1000) |
| Antibody | anti-goat Alexa Fluor 488 (Donkey polyclonal) | Jackson Immuno Research Labs | Cat# 705-545-147 RRID:AB_2336933 | IHC (1:500) |
| Antibody | anti-Rabbit Alexa Fluor 594 (Donkey polyclonal) | Jackson Immuno Research Labs | Cat# 711-585-152RRID:AB_2340621 | IHC(1:500) |
| Antibody | Streptavidin-Alexa Fluor 647 | ThermoFisher Scientific | Cat# S21374RRID:AB_2336066 | IHC (1:2000) |
| Recombinant DNA reagent | AAVrg-pmSyn1-EBFP-Cre ($6\times10^{12}$ GC/mL) | addgene | Cat# 51507-AAVrg | |
| Recombinant DNA reagent | AAV9-EF1a-DIO-hChR2(H134R)-EYFP-WPRE-HGHpA ($2.2\times10^{13}$ GC/mL) | addgene | Cat# 35507-AAVrg | |
| Recombinant DNA reagent | AAV1-CAG-ChR2-Venus-WPRE-SV40 ($8.99\times10^{12}$ GC/mL) | addgene | Cat# 20071-AAV1 | |
| Recombinant DNA reagent | AAV2-EF1a-DIO-hChR2(E123T/T159C)-p2A-eYFP-WPRE | addgene | Cat# 35509-AAV2 | |
| Software, algorithm | pClamp 10 | Molecular Devices | RRID:SCR_011323 | |
| Software, algorithm | Igor Pro 8 | WaveMetrics | RRID:SCR_000325 | |
| Software, algorithm | NeuroMatic | *Rothman and Silver, 2018*; DOI:10.3389/fninf.2018.00014 | RRID:SCR_004186 | |
| Software, algorithm | Axograph | Axograph | RRID:SCR_014284 | |
| software, algorithm | Excel | Microsoft | RRID:SCR_016137 | |
| Software, algorithm | FIJI | Fiji.sc | RRID:SCR_002285 | |
| Software, algorithm | Adobe Illustrator CS2 | Adobe | RRID:SCR_010279 | |

## Animals

Transgenic mice of both sexes expressing Cre recombinase under the endogenous choline acetyltransferase promoter (ChAT-IRES-Cre; Jackson Labs 006410) (*Rossi et al., 2011*) were crossed with a tdTomato reporter line (Ai9(RCL-tdT); Jackson Labs 007909) to generate mice expressing tdTomato in cholinergic neurons (referred to as ChAT-Cre/tdTomato). A small fraction of ChAT-Cre/tdTomato mice exhibit ectopic expression of Cre recombinase, which labels vasculature and astrocytes (https://www.jax.org/strain/006410). When ectopic expression was observed, the slices were not used for experimental data. Mouse lines were maintained in an animal facility managed by the Department of Comparative Medicine at Oregon Health and Science University. All procedures were approved by the Oregon Health and Science University's Institutional Animal Care and Use Committee and met the recommendations of the Society for Neuroscience.

## Immunohistochemistry and imaging

Mice were deeply anesthetized with isoflurane and then perfused through the heart with 0.1 M phosphate buffered saline (PBS), pH 7.4, 33°C, followed by ice-cold 4% paraformaldehyde in 0.1 M PBS using a peristaltic pump. Brains were surgically extracted and incubated with 4% paraformaldehyde in 0.1 M PBS overnight at 4°C. Brains were washed in 0.1 M PBS three times, 10 min per wash, and then 50 μm sections were made on a vibratome (Leica, VT1000S) and saved as floating sections in 0.1 M PBS. To visualize the cells that were filled with biocytin during whole-cell recording, 300 μM acute brain slices were fixed with 4% paraformaldehyde in 0.1 M PBS overnight at 4°C. Sections used for antibody labeling were permeabilized and blocked in 2% bovine serum albumin, 2% fish gelatin, and 0.2% Triton X-100 in 0.1 M PBS for 2 hr at room temperature on a 2-D rocker. Sections were then incubated in primary antibodies for 2 days at 4°C on a 2-D rocker. Sections were washed in 0.1 M PBS three times, 10 min per wash, and then incubated in secondary antibodies and streptavidin-conjugated fluorophores for 2 days at 4°C on a 2-D rocker. See 'Key Resources' table for a full list of antibodies and reagents used. Sections were washed in 0.1 M PBS three times, 10 min each wash, followed by incubation in 4% paraformaldehyde in 0.1 M PBS for 30 min. Some brain sections with high fluorophore expression were not enhanced with antibody labeling to reduce background. All sections were mounted on microscope slides and coverslipped with Fluoromount-G (Southern-Biotech) mounting medium, and then sealed with clear nail polish. All images of histological sections were acquired on a Zeiss LSM780 confocal microscope system. Images were processed for contrast, brightness, and gamma using Fiji (*Schindelin et al., 2012*).

## Acute brain slice preparation

Mice were deeply anesthetized with isoflurane and decapitated. The brain was rapidly extracted while submerged in warm (40°C) artificial cerebral spinal fluid (aCSF) containing (in mM) 130 NaCl, 2.1 KCl, 1.2 $KH_2PO_4$, 3 Na-4-(2-hydroxyethyl)-1-piperazineethanesulfonic acid (HEPES), 11 glucose, 20 $NaHCO_3$, 1 $MgSO_4$, and 1.7 $CaCl_2$, bubbled with 5% $CO_2$/95% $O_2$. Parasagittal and coronal sections of the brain containing the superior olive, cochlear nucleus, or inferior colliculus were cut at 300 μm with a vibratome (VT1200S, Leica, or 7000smz-2, Campden) in warm aCSF. Throughout sectioning, brain slices were collected and stored in aCSF at 31°C. When sectioning was completed, slices were incubated for an additional 30 min at 31°C, followed by storage at room temperature, ~23°C.

## Electrophysiology

Acute brain slices were transferred to a recording chamber and submerged in aCSF. Slices were anchored to the chamber using a platinum harp with nylon threads and placed on a fixed-stage microscope (Axioskop 2 FS Plus, Zeiss). The recording chamber was perfused with aCSF at 3 ml/min and maintained at 31–33°C with an in-line heater (TC-344A; Warner Instrument Corp). Neurons in each slice were viewed using full-field fluorescence with a white-light LED attached to the epifluorescence port of the microscope that was passed through a tdTomato filter set with a X40 water-immersion objective (Zeiss) and a digital camera (Retiga ELECTRO; QImaging). In slices from ChAT-Cre/tdTomato mice, MOC neurons were identified in the VNTB by their tdTomato fluorescence and morphology. Borosilicate glass capillaries (OD 1.5 mm; World Precision Instruments) were pulled on a P-97 Flaming/Brown micropipette puller (Sutter) to a tip resistance of 1–5 MΩ. All whole-cell current-clamp experiments were conducted with an internal pipette solution containing (in mM) 113 K-gluconate, 2.75 $MgCl_2$, 1.75 $MgSO_4$, 9 HEPES, 0.1 ethylene glycol tetraacetic acid (EGTA), 14 tris-phosphocreatine, 0.3 tris-GTP, 4 $Na_2$-ATP, pH adjusted to 7.2 with KOH, and osmolality adjusted to 290 mOsm with sucrose. Whole-cell voltage-clamp experiments were conducted using a K-gluconate-based pipette solution or a cesium-based pipette solution containing (in mM) 103 CsCl, 10 tetraethylammonium chloride (TEA-Cl), 3.5 N-ethyllidocaine chloride (QX-314-Cl), 2.75 $MgCl_2$, 1.74 $MgSO_4$, 9 HEPES, 0.1 EGTA, 0.1 spermine, 14 tris-phosphocreatine, 0.3 tris-GTP, 4 $Na_2$-ATP, with pH adjusted to 7.2 with CsOH, and osmolality adjusted to 290 mOsm with sucrose. All IV-relation experiments used a cesium-based pipette solution. Polyamine-free cesium-based pipette solutions omitted spermine. Reported voltages were corrected for their liquid junction potential: −12.4 mV for K-gluconate-based pipette solution and −2.8 mV for cesium-based pipette solution. Loose-patch recordings were conducted with aCSF as the pipette solution. In some experiments, 0.1% biocytin

(B1592, Thermo Fisher Scientific) was added to the pipette solution for post-hoc identification of MOC neurons. Whole-cell recordings were amplified (5X gain), low-pass filtered (14 kHz Bessel, Multiclamp 700B; Molecular Devices), and digitized using pClamp software (50–80 kHz, Digidata 1440A; Molecular Devices). Series resistance compensation was set to 60% correction and prediction with a bandwidth of 1.02 kHz. The majority of pipettes used for voltage clamp were wrapped with Parafilm M (Bemis) to reduce pipette capacitance. Cells were voltage-clamped at −62.8 mV unless noted otherwise. The average uncompensated series resistance ($R_s$) when patched onto a neuron was 14.5 ± 0.9 MΩ. 1 mM glutamate in aCSF was puffed onto cells from a patch pipette attached to a Picospritzer II (Parker). The puff pressure was adjusted between 5 and 10 psi for 2–15 ms duration to achieve stable glutamate-evoked currents. The baseline current (50–200 ms) was subtracted from the data in all example current traces unless noted otherwise. ChR2 was activated using 2 ms flashes of light through a green fluorescent protein (GFP) filter set from a 470 nm LED attached to the epifluorescence port of the microscope. Light stimulation was made through a X40 water immersion objective (Zeiss). At some synapses, ChR2 stimulation can exhibit artificial synaptic depression (*Jackman et al., 2014*). To confirm that light-evoked short-term plasticity observed from activation of MOC neuron inputs was not an artifact of ChR2 stimulation at presynaptic boutons (i.e., action potential broadening, increasing the probability of vesicle release), light stimulation was compared over input axons and MOC neuron somata (*Figure 7—figure supplement 1*). In sagittal sections, moving the objective lens away from the recorded neuron and toward the IC in 230 µm steps delayed the onset of light-evoked EPSCs. No EPSC could be evoked when light stimulation was directly ventral to the recorded neuron where there was an absence of brain tissue, confirming that light stimulation was confined to the location of the objective lens. A plot of change in EPSC delay over camera position was best fit with a linear equation, and the axon conduction velocity was calculated to be 0.571 m/s (*Figure 7—figure supplement 1C*). Similar short-term plasticity was observed with both axonal and somatic stimulation (*Figure 7—figure supplement 1D*).

## Miniature EPSC analysis

Miniature EPSCs were recorded in the presence of 1 µM TTX to block spontaneous spike-driven events, 0.5 µM strychnine and 10 µM SR95531 to block inhibitory receptors, and 10 µM MK-801 to block NMDA receptors. Spontaneous miniature events were detected using a template search function in AxoGraph (1.7.4) from continuously collected data that were stable for more than 3 min. Events were captured and aligned by their onset, and then the average amplitude, time course, and IEI was calculated. Events that appeared artificial or contained multiple EPSCs were rejected by eye. The decay of average mEPSC data was analyzed in Igor Pro 8 (WaveMetrics) and fit with a double exponential equation, $I(t) = A_{fast} \exp\left(\frac{-t}{\tau_{fast}}\right) + A_{slow} \exp\left(\frac{-t}{\tau_{slow}}\right)$, where $I(t)$ is the current as a function of time, $\tau_{fast}$ and $\tau_{slow}$ reveal fast and slow decay time constants, and $A_{fast}$ and $A_{slow}$ are their relative amplitudes.

## Conductance clamp

To accurately record the membrane voltage while simultaneously injecting conductance waveforms, individual MOC neurons were patched simultaneously with two recording electrodes in whole-cell configuration, both containing K-gluconate-based pipette solution (*Figure 8—figure supplement 1A*). One electrode served as a voltage follower while the other injected current, thereby avoiding the possibility of distortion of fast waveforms by voltage drop across the series resistance or of capacity transients. Conductance-clamp experiments were recorded in the presence of aCSF containing 0.5 µM strychnine, 10 µM SR95531, 10 µM MK-801, and 5 µM 2,3-dioxo-6-nitro-1,2,3,4-tetrahydrobenzodf]quinoxaline-7-sulfonamide (NBQX) to block all major inhibitory and excitatory inputs. Simulated EPSGs were injected using an analog conductance injection circuit, Synaptic Module 1 (SM-1) (Cambridge Conductance), driven by a digital computer. The SM-1 unit was set to rectifying mode and $E_{rev}$ was set to +10 mV, to simulate MOC neuron CP-AMPARs (*Figures 3F* and *5E*).

Conductance waveforms were created using Igor Pro 8 and modeled using physiological data. Whole-cell recordings were made from MOC neurons in the presence of 0.5 nM strychnine, 10 µM SR95531, and 10 µM MK-801 to isolate AMPAR-mediated EPSCs. ChR2-positive IC or VCN input was minimally stimulated (~50% chance of evoking an EPSC) using a Lambda TLED light source (Sutter) to reveal unitary responses. Unitary responses were measured to have a maximum conductance

($G_{max}$) of 0.40 ± 0.01 nS for IC inputs ($N$ = 3) and 0.46 ± 0.06 nS for VCN inputs ($N$ = 3). Thus, simulated unitary EPSGs were set to a $G_{max}$ of 0.40 nS for synapses modeling short-term facilitation (IC input) or 0.46 nS for those modeling short-term depression (VCN input) (*Figure 8* and *9*). The unitary EPSG waveform, $EPSG(t) = \left(1 - e^{\frac{-t}{\tau_{rise}}}\right) \times \left(e^{\frac{-t}{\tau_{decay}}}\right)$, was based on a fit to averaged EPSCs from IC and VCN, with $\tau_{rise}$ = 0.27 ms and $\tau_{decay}$ = 1.9 ms. Timing and frequency of EPSGs were convolved to action potential timing from T-stellate cells in response to repeated 500-ms current injections (*Figure 8—figure supplement 1B and C*), and each repetition (trial) was considered an input (*Figures 8A* and *9A*). T-stellate cells were identified by virally mediated retrograde labeling (AAVrg-pmSyn1-EBFP-Cre) (*Figure 4*) from the contralateral IC in Ai9(RCL-tdT) mice. For each individual input, short-term plasticity was simulated in a frequency-invariant manner by weighting $G_{max}$ of unitary EPSGs according to exponential fits of normalized physiological data (*Figure 7D*). A caveat is that the measured profile of plasticity was extrapolated to presynaptic rates beyond the range studied, where frequency-dependent plasticity could be a factor. For short-term facilitation, $EPSG_0 = G_{max}$ and $EPSG_n = G_{max}\left(Fac_{max} + Ae^{\frac{-n}{\tau}}\right)$, where $Fac_{max}$ = 2.43, $\tau$ = 12.9, and A = -1.42. For short-term depression, $EPSG_0 = G_{max}$ and $EPSG_n = G_{max}\left(Dep_{max} + A_1 e^{\frac{-n}{\tau_1}} + A_2 e^{\frac{-n}{\tau_2}}\right)$, where $Dep_{max}$ = 0.309, $\tau_1$ = 0.771, $A_1$ = 0.443, $\tau_2$ = 4.10, and $A_2$ = 0.248.

## Stereotactic injections

Glass capillaries (WireTrol II; Drummond Scientific) were pulled on a P-97 Flaming/Brown micropipette puller (Sutter) and beveled to 45-degree angle with a tip diameter of 30–40 µm using a diamond lapping disc (0.5 µm grit, 3M). Mice (P22-24) were anesthetized with isoflurane (5% induction, 1.5–2% maintenance) and secured in a small stereotaxic frame (David Kopf). While mice were under isoflurane anesthesia, viral injections were made with a single-axis manipulator (MO-10; Narishige) and pipette vice (Ronal) attached to a triple-axis motorized manipulator (IVM Triple; Scientifica). After application of 10% povidone iodine, the scalp was cut, and the head was leveled using bregma and lambda. The lateral-medial axis was leveled by focusing a X10 objective 2 mm lateral from lambda to be in the same focal plane on the left and right skull. The location of IC was visually detected after removing a 1 mm² unilateral section of occipital bone directly caudal to the lambdoid suture and was pressure injected at a depth of 1 mm. After removing a 1 mm caudal by 2 mm lateral unilateral section of occipital bone caudal to the lambdoid suture, the VCN was located by stereotactic coordinates (0.7 mm lateral, 0.95 mm rostral, and 4.0 mm depth) starting from the surface junction point of the IC, cerebellar lobule IV-V, and simple lobule, which is often marked by a Y-shaped branch from the transverse sinus. Post-injection, the incision was closed with nylon sutures. For analgesia, mice were subcutaneously administered 1 mg/kg meloxicam during surgery and once a day for the 3 following days. Experiments were conducted 1–3 weeks post-surgery.

## Posterior semi-circular canal injections

Our protocol was adapted from *Suzuki et al., 2017*, who developed a procedure to deliver viral vectors to the cochlea via the posterior semi-circular canal (PSCC), which minimizes auditory system damage. Briefly, mice were anesthetized and secured to a stereotaxic frame in a manner identical to stereotactic injections, and then rotated 90 degrees onto their side. A small post-auricular incision was made and muscle tissue overlying the temporal bone was dissected to reveal the bony wall of the PSCC. A small hole was made in the PSCC using a 26-gauge hypodermic needle (Kendall), and lymphatic fluid was allowed to drain for 5 min. The tip of a small polyethylene tube (PE-10) attached to a pipette vice (Ronal) containing CTB (0.5% in 0.05 M Tris, 0.2 M NaCl, 0.001 M NaEDTA, 0.003 M NaN3, pH 7.5) was placed into the PSCC oriented toward the ampulla, and sealed with fragments of muscle and cyanoacrylate glue (3M Vetbond Tissue Adhesive). 1–2 microliters of CTB was injected into the PSCC, and the polyethylene tube was left in place for an additional 5 min. After removing the polyethylene tube, the hole was plugged with small pieces of muscle and covered with cyanoacrylate glue. The skin was closed with nylon sutures and mice were perfused for histochemistry 1–5 days later.

## Experimental design and statistical analysis

Electrophysiological traces were analyzed with pClamp 10.7 (Molecular Devices) and IGOR Pro 8 (Wavemetrics) using the NeuroMatic 3.0 package (*Rothman and Silver, 2018*). Miniature events were analyzed with Axograph 1.7.4 (*Clements and Bekkers, 1997*) and IGOR Pro 8. Averages are represented as mean ± SEM. Statistical analysis was conducted with IGOR Pro 8, and the significance between group means was examined using a two-way analysis of variance (ANOVA) test with a post-hoc Tukey test to identify means that significantly differed. Two-tailed Student's *t*-test was used for comparison between two means. The significance threshold was set at $p < 0.05$ for all statistical tests. Figures were created with IGOR Pro eight and Adobe Illustrator (CS2).

## Acknowledgements

We would like to thank Ruby Larisch, Jennifer Goldsmith, and Sean Elkins for help with mouse husbandry and genotyping, Stefanie Kaech Petrie and Aurelie Snyder at the OHSU Advanced Light Microscopy Core for their help with imaging (Core supported by P30 NS0618000 to Sue Aicher), and NIH F31 DC016226 (GER) and DC004450 (LOT). Gabriel Romero was a Howard Hughes Medical Institute Gilliam fellow.

## Additional information

### Funding

| Funder | Grant reference number | Author |
| --- | --- | --- |
| National Institutes of Health | DC016226 | Gabriel E Romero |
| Howard Hughes Medical Institute | Gilliam Fellowship | Gabriel E Romero |
| National Institutes of Health | DC004450 | Laurence O Trussell |

The funders had no role in study design, data collection and interpretation, or the decision to submit the work for publication.

### Author contributions

Gabriel E Romero, Laurence O Trussell, Conceptualization, Data curation, Software, Formal analysis, Supervision, Funding acquisition, Validation, Investigation, Visualization, Methodology, Writing - original draft, Writing - review and editing

### Author ORCIDs

Gabriel E Romero (iD) https://orcid.org/0000-0002-6086-5546
Laurence O Trussell (iD) https://orcid.org/0000-0003-1171-2356

### Ethics

Animal experimentation: Animal experimentation: All experiments were performed under the approval of the institutional animal care and use committee (IACUC) of Oregon Health and Science University, assurance #A3304-01.

### Decision letter and Author response

Decision letter https://doi.org/10.7554/eLife.66396.sa1
Author response https://doi.org/10.7554/eLife.66396.sa2

## Additional files

### Supplementary files

- Transparent reporting form

## Data availability
All data are provided the manuscript.

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
