## [Decision Letter]

**Acceptance summary:**

The manuscript by Romero and Trussell elegantly investigates the inputs to cholinergic neurons in the ventral nucleus of the trapezoid body, which provides the primary efferent projections to the outer hair cells of the cochlea (the "MOC" system). This efferent control system is thought to play a role in adjusting the cochlear gain, improving the detection of signals in noise, and potentially providing some protection from acoustic trauma. The authors use a combination of genetically modified mice, virus tracing and cell-type-specific insertion of channelrhodposins, and electrophysiology methods to identify and characterize the synaptic inputs to these cells.

**Decision letter after peer review:**

Thank you for submitting your article "Distinct forms of synaptic plasticity during ascending vs descending control of medial olivocochlear efferent neurons" for consideration by *eLife*. Your article has been reviewed by 2 peer reviewers, and the evaluation has been overseen by a Reviewing Editor and Andrew King as the Senior Editor. The following individuals involved in review of your submission have agreed to reveal their identity: Conny Kopp-Scheinpflug (Reviewer #3).

The reviewers have discussed their reviews with one another, and the Reviewing Editor has drafted this to help you prepare a revised submission. We see some issues (detailed below) but agree that your primary findings are significant. We suggest the following essential revisions, to modify claims and make clear that some conclusions will need supporting data in the future.

We are of the opinion that an understanding of the impact of inhibition would provide a more complete picture, but acknowledge that this would require a new study. We also think the application of faster opsins would be informative. In both cases, however, this would require new experiments. We suggest instead that you discuss these caveats.

Essential Revisions:

1) Please adjust some of the claims of the paper to reflect the limitations in your interpretation. Specifically, to fully support the claims from the dynamic clamp experiments, you should acknowledge that substantial additional work would be needed. Suggestions include addressing inhibition, identifying the IC neurons that are the source for the descending excitatory input, and extending the frequency range in the optogenetic component.

2) The dynamic clamp extrapolation in the frequency domain should be more clearly acknowledged and discussed. More detail is available in the full reviews below.

*Reviewer #2:*

The manuscript by Romero and Trussell elegantly investigates the inputs to cholinergic neurons in the ventral nucleus of the trapezoid body, which provides the primary efferent projections to the outer hair cells of the cochlea (the "MOC" system). This efferent control system is thought to play a role in adjusting the cochlear gain, improving the detection of signals in noise, and potentially providing some protection from acoustic trauma. The authors use a combination of genetically modified mice, virus tracing and cell-type-specific insertion of channelrhodposins, and electrophysiology methods to identify and characterize the synaptic inputs to these cells.

They focus on two sources. The first source is ascending input from cochlear nucleus neurons, T-Stellate cells, which are identifiable at a cellular and molecular-genetic level. The second source is descending input from unidentified neurons of the inferior colliculus. The work convincingly demonstrates several features of these excitatory inputs. Both inputs drive rapidly desensitizing, rectifying, and calcium-permeable AMPA receptors. However, the synapses show opposing release dynamics. The ascending synapses exhibit depression at the two frequencies tested, 20 and 50 Hz, whereas the descending synapses show facilitation at these frequencies. Somewhat surprisingly, the recovery from depression and facilitation for the two inputs follow similar time courses. Some of the potential concerns regarding the limitations of optical activation of ChR2-expressing axons and terminals are addressed. These experiments are thoughtfully performed and presented, and provide the first detailed characterization of these two excitatory synaptic inputs to the MOC neurons. These results suggest that the ascending inputs provide sensory information, whereas the descending inputs offer a slower, modulatory control over the efferent neurons. A limitation is that the exact cellular source of the descending inputs is not known.

The authors extend the characterization of the two input sources with dynamic clamp experiments that compare how the firing of the MOC neurons is affected when the inputs are simulated both separately and simultaneously, including at a higher firing rate. The experiments attempt to reveal distinct functional roles for the ascending and descending inputs. The authors use in vitro responses to current pulses recorded from T-Stellate neurons for spike timing. The experiments are technically challenging (two-electrodes on one target cell for dynamic clamp) and could provide valuable information. The results suggest that the combination of depressing and facilitating inputs better supports sustained firing and enhances the dynamic range of firing rates with increasing input. The combined inputs also result in a relatively constant, short-latency to the first spike compared to when tested independently. However, there are several limitations to these experiments. The most significant limitation is the choice of input stimulus patterns, which do not accurately reflect the adapting responses of the afferent neurons to acoustic stimuli (and in the case of the IC input, we do not know what the response patterns should be as the source neurons have not been characterized). A second limitation is that the synaptic dynamics are treated as independent of frequency. This assumption conflicts with data shown in the paper, where the depression and facilitation rates are frequency-dependent in a manner typical of many previously studied synapses. Further, some frequencies used in the dynamic clamp experiments (110, 180 Hz), although reasonable based on in vivo data, were outside the range studied optogenetically (20, 50 Hz), so it is not clear that the same synaptic dynamics apply. An additional concern is that the dynamic clamp method only simulates somatic inputs, so dendritic filtering effects are lost (Figure 3 shows that putative terminals from the VCN appear in the neuropil around the cells in the VNTB; the preponderance of somatic versus dendritic targeting is not clear). The dynamic clamp experiments are essentially thought experiments that suggest possible roles for the distinct synaptic dynamics of the ascending and descending inputs and incorporate a number of the non-linear factors inherent in the biology of the system. However, they also have significant limitations.

Overall, the manuscript generally provides well-documented new and intriguing observations regarding the pathways driving the MOC neurons. Perhaps the most important result is the characterization of the descending input to the MOC neurons and the suggestion that it plays an essential role in how the MOC neurons operate. There are many open questions that are raised by the observations in this paper, including the types, sensory responses, and other inputs of the inferior colliculus neurons that project to the MOC neurons. To their credit, the authors do carefully address several caveats regarding the interpretations and limits of the approaches in the discussion and provide an appropriately cautious interpretation of their results.

Overall in the development of the background and in the discussion, the system is viewed from the standpoint of experiments in cats, and to some extent rats, and a lesser extent, mice. Although there may be some equivalences, and these are excellent strong motivations for your work, I would be cautious about making arguments from other species when interpreting specific points. This may require a deeper dive into the literature. Looking particularly at line 503-504.

Although the authors emphasize the linearity of the FI relationship this is really only clear up to about 1 nA, and even then a few cells show the more typical roll-over of spike rate. Therefore, it would be appropriate to temper the "linearity" conclusions.

Dynamic Clamp Experiments:

Although these experiments are well motivated and some aspects are technically well done, I had concerns about the specific parameters used for the synaptic facilitation and depression, as indicated in the Public Review. To clarify, a better model of the synaptic dynamics, incorporating a wider frequency range appropriate to the stimuli used, would provide stronger support for the conclusions of these experiments. Clearly there is a difference in the depression and facilitation rates between 20 and 50 Hz, and I would expect that there would be even greater differences at lower and higher frequencies. The best approach would be to do a full assessment of the dynamics over a wider frequency range (10-400 Hz), including post-train recovery, and then fitting these to a single model (e.g., Tsodyks or Dittman et al.,) to use in the dynamic clamp. Then the facilitation and depression rates for each synapse would be tailored to the full frequency range and the use of frequencies outside of the experimentally-measured range would not be an issue. On the other hand, it may not be easy to obtain the required data rigorously from an experiment at present. Electrical stimulation would likely be problematic, but optogenetics would require using faster opsins, if the appropriate controls could be done.

The recording from unidentified ChR2 positive neurons in the IC (Figure 5) provides some evidence that the cells can follow 20 Hz stimulation, but does not provide sufficient characterization of the properties of those cells. Excitatory IC neurons exhibit a diversity of intrinsic physiology, responses to acoustic stimuli, and molecular identity. An experiment to identify which specific IC neuron class(es) project to the MOC neurons may be difficult and would take some time to work out and is beyond the scope of the present work. However, it is unclear that those cells in the IC that are labelled and were recorded in slices are necessarily the same ones that project to the MOC neurons. This presents a weakness that needs to be considered when interpreting the dynamic clamp data. The dynamic clamp experiments assume that the inputs from the IC fire regularly, without adaptation, but this may not be the case. This needs to be carefully discussed in terms of the importance of synaptic facilitation versus depression.

Figure 1 panel F: I found this presentation confusing (because CTB-positive refers to CTB+ChAT-td cells; theoretically there could be CTB-positive, ChAT-td negative cells). Perhaps clarifying the labeling would help.

line 333: I am not sure what "fully engage" means in this context.

line 357: It is also possible that the facilitation is a compensation for firing rate adaptation, such that over time the synaptic strength is relatively constant. In that case, the cells may be tonic firing, but show strong adaptation. See also comment on Figure 5.

line 405: As the IC is at least one synapse later than the VCN input, and has a longer pathway to travel (both ascending and descending), it would be expected that it is also delayed. In the experiments where two inputs are combined, was this delay taken into account?

line 373: "small number" : be more specific.

line 486: This needs a few citations. There are examples in the auditory system that may be relevant (or even lead to an opposing conclusion), but at least some backup for the statement should be present so readers know what you are thinking about.

Line 530: The speed of the receptor kinetics might not be the important observation here given the relatively slow output of the MOC system. Did you consider that the receptor-associated calcium influx might be important? This could be discussed.

line 572: "Insertion" seems the wrong word here. Perhaps "Activation"?

Line 670: this should include "uncompensated" in the description for clarity.

line 673: "baselined" ? You mean the baseline current (define time frame) was subtracted from the data. Please reword.

line 734-735: The tau values and amplitudes have units; I assume these are in sec and nA respectively, but they should be specified. The meaning of "n" and "n0" is not defined.

Figure 4Ai shows T-stellate cells projecting to the DCN in the schematic. However, 4Ei does not show any visible terminals in the DCN. Also, the lack of terminals in the DCN is not mentioned in the text (lines 216-218). At face value, this suggests that those T-stellate cells that project to the DCN are not the same population that projects to the IC. However, this would contradict data in the cat (Adams, 1983), which shows that double-labeling of a substantial population of "multipolar" cells in the VCN after combined IC and DCN injections. On the other hand, it is not clear from Oertel's reconstructions that all T-stellate cells have a collateral to the DCN (though clearly limitations on that data). Those of Smith and Rhode's (89) cat paper, where axons could be traced, do. Perhaps some comment on this would be appropriate.

Figure 5: The sample size for the loose-patch cell attached recordings in the IC is not given.

*Reviewer #3:*

In addition to the main conclusion, the manuscript includes a general description of MOC neuron physiology in mature mice using a reporter line to visualize cholinergic neurons. These descriptions could serve as markers to identify MOC neurons in future experiments even without the reporter line.

Overall, the findings are very interesting as they question the MOC system as a reflex arc, given that the ascending input can be overwritten or at least modified by descending commands. The manuscript is prepared at the highest standard. The experiments are well constructed and of excellent quality. Especially the use of an intersectional AAV guided expression of channelrhodopsin to study the origin and function of MOC inputs and performing 2-electrode dynamic clamp to examine their interplay is very elegant.

Including the role of the known inhibitory input to the MOC neurons into the evaluation of the MOC output would add an important piece to our understanding of this efferent circuit.

I have only a few suggestions that could further improve the functional interpretation.

The manuscript focusses strictly on glutamatergic activation of AMPA receptors. This raises two questions: First, do MOC cells express NMDA receptors or can we neglect them? The second and major question is: How is inhibition integrated into generating the MOC output? The conductance clamp experiment is very good and I am not suggesting to do a 3-electrode recording, but it would be very useful to use the published (Torres Cadenas et al., 2020) information on the inhibitory input to the MOC cells to test their influence in the conductance clamp experiment. This could be done testing one excitatory input at a time in combination with the inhibition. Especially with respect to the apparent discrepancy between the high firing rates suggested by the current injection protocols and the rather low firing rates in vivo. As discussed by the authors, it could be the effect of anesthesia, but it could also be the impact of inhibition. Only then, a good prediction of MOC output should be possible.

[Editors' note: further revisions were suggested prior to acceptance, as described below.]

Thank you for resubmitting your work entitled "Distinct forms of synaptic plasticity during ascending vs descending control of medial olivocochlear efferent neurons" for further consideration by *eLife*. Your revised article has been evaluated by Andrew King (Senior Editor) and a Reviewing Editor following consultation with the original reviewers. Here is a summary of our evaluation of the revised version.

The manuscript has been improved, but some remaining issues have been raised by reviewer #2 that need to be addressed, as outlined below:

1. Regarding: "To these points we would like to note that the T-stellate cell firing in vivo does not mirror the adapting responses of the afferent neurons. Rather they show the characteristic 'chopper' responses described by Sachs and colleagues years ago. Moreover, we found that the chopper response of the T-stellate cell is recapitulated by simply applying current injection stimuli repeated over and over."

I think that the authors and I have somewhat different readings of the past literature here. "Choppers" in the VCN exist as a continuum of discharge patterns in cat (Godfrey et al., 1976; Bourk, 1976; Typlt et al., 2012), some of which have pronounced adaptation during short tone bursts (Bourk, 1976; Young et al., J. Neurophys. 1988; Blackburn and Sachs, J. Neurophysiol. 1989; Winter and Palmer, Hearing Res. 1990 Figure 7; guinea pig; Rhode and Smith, 1986, Figure 17 at 40dB SPL). Similar adaptation is evident in mouse (see Roos and May, Hearing Res. 2012 Figures 5 and 6). Quantification of regularity in responses to longer tone bursts (seen in Godfrey et al., 1975) is generally lacking in the literature. The point is that in some "choppers", there is rate adaptation, driven by a combination of the multiple phases of adaptation in the auditory nerve rate, and by inhibition that increases over time. I agree that most stellate cells in the CN show little adaptation in slice recordings in response to direct current injection, as indicated by the authors, but a few do (Figure 3, Manis et al., 2019).

It is ok for the authors to do the experiment in the way that they did, but they also should not ignore the literature that indicates that in responses to tone bursts in vivo, adaptation over the time scale of their stimulus sets does occur in some potential afferent cell populations.

2. Regarding: "A second limitation is that the synaptic dynamics are treated as independent of frequency. This assumption conflicts with data shown in the paper, where the depression and facilitation rates are frequency-dependent in a manner typical of many previously studied synapses." Response: "We respectfully disagree with this point. Our figure 7D clearly shows profiles of depression or facilitation that are identical after nearly tripling the stimulation rate, from 20 to 50 Hz."

This depends on the assumptions that are made regarding the dynamic behavior of synapses when challenged with different rates of spikes, and on the lack of data at these synapses regarding the dynamic behavior across a wider range of stimulus frequencies. The data shown in the paper at two frequencies are plotted as a function of spike number during the train, which translates into two different depression rates (e.g., time constants). Synapses are inherently non-linear, so that measurements made at two relatively low frequencies (20, 50 Hz) should not be extrapolated to higher frequencies (180 Hz) without acknowledging this limitation. The demonstration by Wang and Kaczmarek (1998) of the engagement of a frequency-dependent recovery from depression at high rates is a clear example of this point (and one that has been shown at other synapses). The authors should recognize in the text that they have used a simplified situation in which rate-dependent factors that may accrue outside the range that they were able to study are not considered. I do not think it much affects their conclusions as far as they go, but it is just a caveat worth mentioning.

3. The issue with dynamic clamp is a real one, but is also second-order. It also cannot be properly evaluated because the roles of dendritic ion channels in these cells are not known, and the consequences of dendritic conductance changes (e.g., synaptic inputs) and somatic conductance changes imposed by dynamic clamp are not exactly the same, although they are qualitatively very similar. I accept that you do the experiments that you can; however I do wish that the authors would acknowledge that there are limitations to this approach.

---

## [Author Response]

Essential Revisions:1) Please adjust some of the claims of the paper to reflect the limitations in your interpretation. Specifically, to fully support the claims from the dynamic clamp experiments, you should acknowledge that substantial additional work would be needed. Suggestions include addressing inhibition, identifying the IC neurons that are the source for the descending excitatory input, and extending the frequency range in the optogenetic component.

Each of these points has now been addressed in the Discussion, and are highlighted in the revised copy.

2) The dynamic clamp extrapolation in the frequency domain should be more clearly acknowledged and discussed. More detail is available in the full reviews below.

We have responses to these comments in the full reviews.

Reviewer #2:The manuscript by Romero and Trussell elegantly investigates the inputs to cholinergic neurons in the ventral nucleus of the trapezoid body, which provides the primary efferent projections to the outer hair cells of the cochlea (the "MOC" system). This efferent control system is thought to play a role in adjusting the cochlear gain, improving the detection of signals in noise, and potentially providing some protection from acoustic trauma. The authors use a combination of genetically modified mice, virus tracing and cell-type-specific insertion of channelrhodposins, and electrophysiology methods to identify and characterize the synaptic inputs to these cells.They focus on two sources. The first source is ascending input from cochlear nucleus neurons, T-Stellate cells, which are identifiable at a cellular and molecular-genetic level. The second source is descending input from unidentified neurons of the inferior colliculus. The work convincingly demonstrates several features of these excitatory inputs. Both inputs drive rapidly desensitizing, rectifying, and calcium-permeable AMPA receptors. However, the synapses show opposing release dynamics. The ascending synapses exhibit depression at the two frequencies tested, 20 and 50 Hz, whereas the descending synapses show facilitation at these frequencies. Somewhat surprisingly, the recovery from depression and facilitation for the two inputs follow similar time courses. Some of the potential concerns regarding the limitations of optical activation of ChR2-expressing axons and terminals are addressed. These experiments are thoughtfully performed and presented, and provide the first detailed characterization of these two excitatory synaptic inputs to the MOC neurons. These results suggest that the ascending inputs provide sensory information, whereas the descending inputs offer a slower, modulatory control over the efferent neurons. A limitation is that the exact cellular source of the descending inputs is not known.The authors extend the characterization of the two input sources with dynamic clamp experiments that compare how the firing of the MOC neurons is affected when the inputs are simulated both separately and simultaneously, including at a higher firing rate. The experiments attempt to reveal distinct functional roles for the ascending and descending inputs. The authors use in vitro responses to current pulses recorded from T-Stellate neurons for spike timing. The experiments are technically challenging (two-electrodes on one target cell for dynamic clamp) and could provide valuable information. The results suggest that the combination of depressing and facilitating inputs better supports sustained firing and enhances the dynamic range of firing rates with increasing input. The combined inputs also result in a relatively constant, short-latency to the first spike compared to when tested independently. However, there are several limitations to these experiments. The most significant limitation is the choice of input stimulus patterns, which do not accurately reflect the adapting responses of the afferent neurons to acoustic stimuli (and in the case of the IC input, we do not know what the response patterns should be as the source neurons have not been characterized).

To these points we would like to note that the T-stellate cell firing in vivo does not mirror the adapting responses of the afferent neurons. Rather they show the characteristic ‘chopper’ responses described by Sachs and colleagues years ago. Moreover, we found that the chopper response of the T-stellate cell is recapitulated by simply applying current injection stimuli repeated over and over. It was therefore reasonable to use these responses to drive the simulated synaptic conductances. In order to help clarify this point, we have added to Figure 8 Suppl 1 original traces demonstrating the ‘chopping’ pattern of firing in the T-stellate cell used for our stimuli, a pattern well known to reflect the in vivo response profile of these neurons. We do agree that in the case of the IC input, we do not know the response patterns as those neurons have not been characterized (as now pointed out in the text). However, the firing rate of these neurons is not likely to match that of the VCN, as shown in numerous studies. We assumed that the IC input does show some degree of regular firing, simply because their synapses appear uniquely suited to respond to regular firing with facilitation.

A second limitation is that the synaptic dynamics are treated as independent of frequency. This assumption conflicts with data shown in the paper, where the depression and facilitation rates are frequency-dependent in a manner typical of many previously studied synapses.

We respectfully disagree with this point. Our figure 7D clearly shows profiles of depression or facilitation that are identical after nearly tripling the stimulation rate, from 20 to 50 Hz.

Further, some frequencies used in the dynamic clamp experiments (110, 180 Hz), although reasonable based on in vivo data, were outside the range studied optogenetically (20, 50 Hz), so it is not clear that the same synaptic dynamics apply. An additional concern is that the dynamic clamp method only simulates somatic inputs, so dendritic filtering effects are lost (Figure 3 shows that putative terminals from the VCN appear in the neuropil around the cells in the VNTB; the preponderance of somatic versus dendritic targeting is not clear).

It is true that the stimulation mimics only the effects of the filtered EPSPs on spike generation. But since we measured those EPSPs at soma and mimicked them quantitatively in the dynamic clamp, the dendritic filtering effects are in fact preserved, not lost. Again, the key measurement here is spiking, and that is driven by the somatic/AIS depolarizations which we have both measured and mimicked.

Overall in the development of the background and in the discussion, the system is viewed from the standpoint of experiments in cats, and to some extent rats, and a lesser extent, mice. Although there may be some equivalences, and these are excellent strong motivations for your work, I would be cautious about making arguments from other species when interpreting specific points. This may require a deeper dive into the literature. Looking particularly at line 503-504.

Thank you for reminding us of this caution. We have now updated the text (line 535) to specify that bushy cell projections to VNTB were observed in cat.

Although the authors emphasize the linearity of the FI relationship this is really only clear up to about 1 nA, and even then a few cells show the more typical roll-over of spike rate. Therefore, it would be appropriate to temper the "linearity" conclusions.

While it is true that a few cells lose their SR linearity after a 1 to 2 nA square pulse current injection (Figure 2D), stimuli of these intensities are very likely beyond physiological range. MOC neurons have rarely been reported to fire more than 100 Hz in vivo (lines 504-505), while the majority of the responses we reported exceed this rate even at 900 pA (Figure 2C). Additionally, our conductance clamp experiments which simulated more synaptic-like stimuli recapitulated our linearity observation (Figure 8E, and 9D). Also see lines 494-498, where we compare our results to those of previous studies that also observed a linear increase in spike rate with increasing stimulus intensity.

Dynamic Clamp Experiments:Although these experiments are well motivated and some aspects are technically well done, I had concerns about the specific parameters used for the synaptic facilitation and depression, as indicated in the Public Review. To clarify, a better model of the synaptic dynamics, incorporating a wider frequency range appropriate to the stimuli used, would provide stronger support for the conclusions of these experiments. Clearly there is a difference in the depression and facilitation rates between 20 and 50 Hz, and I would expect that there would be even greater differences at lower and higher frequencies.

Respectfully, our observation (Figure 7) was that facilitation and depression did not differ in their recovery, nor in their corresponding plasticity index between 20 and 50 Hz. Figure 7D demonstrates that the increase in EPSC amplitude was stimulus # dependent at these rates, and not dependent on the rate itself. Therefore, we felt it appropriate to extrapolate our observations to higher simulated rates. Additionally, our goal was to determine how MOC neurons would respond to inputs with similar properties to those we observed, but at higher rates, as this was not possible with our ChR2 construct. We highlighted this in the text (lines 347-350).

The best approach would be to do a full assessment of the dynamics over a wider frequency range (10-400 Hz), including post-train recovery, and then fitting these to a single model (e.g., Tsodyks or Dittman et al.,) to use in the dynamic clamp. Then the facilitation and depression rates for each synapse would be tailored to the full frequency range and the use of frequencies outside of the experimentally-measured range would not be an issue. On the other hand, it may not be easy to obtain the required data rigorously from an experiment at present. Electrical stimulation would likely be problematic, but optogenetics would require using faster opsins, if the appropriate controls could be done.

We agree with the sentiment of this statement and in fact did attempt experiments with alternative opsins (ChETA and Chronos). However, even with the same AAV serotype and promoter, these opsins were consistently observed in MOC neurons postsynaptic to our infected inputs – in other words, the opsin expression was aberrant. Additionally, when using AAVs expressing these opsins to transduce IC neurons, we often observed that T-stellate cells in the VCN were retrogradely infected—something that never happened with AAV1-CAG-ChR2-Venus. Understanding why this transsynaptic and retrograde expression occurred was beyond the aim of our investigation and use of these constructs would unfortunately have confounded our results.

The recording from unidentified ChR2 positive neurons in the IC (Figure 5) provides some evidence that the cells can follow 20 Hz stimulation, but does not provide sufficient characterization of the properties of those cells. Excitatory IC neurons exhibit a diversity of intrinsic physiology, responses to acoustic stimuli, and molecular identity. An experiment to identify which specific IC neuron class(es) project to the MOC neurons may be difficult and would take some time to work out and is beyond the scope of the present work. However, it is unclear that those cells in the IC that are labelled and were recorded in slices are necessarily the same ones that project to the MOC neurons. This presents a weakness that needs to be considered when interpreting the dynamic clamp data. The dynamic clamp experiments assume that the inputs from the IC fire regularly, without adaptation, but this may not be the case. This needs to be carefully discussed in terms of the importance of synaptic facilitation versus depression.

It is true that the specific class of IC projection to MOC neurons is unknown and discovering the identity of this input would be a worthwhile follow-up experiment. We have updated lines 371-375 to clarify that the tonic firing simulated in our dynamic clamp experiments was an assumption due to a lack of knowledge about these descending inputs.

Figure 1 panel F: I found this presentation confusing (because CTB-positive refers to CTB+ChAT-td cells; theoretically there could be CTB-positive, ChAT-td negative cells). Perhaps clarifying the labeling would help.

Thank you, we updated Figure 1F to say “CTB and ChAT” positive. Surprisingly, we never encountered a CTB-positive, ChAT-td negative VNTB neuron.

line 333: I am not sure what "fully engage" means in this context.

Changed to “utilize the wide firing range”.

line 357: It is also possible that the facilitation is a compensation for firing rate adaptation, such that over time the synaptic strength is relatively constant. In that case, the cells may be tonic firing, but show strong adaptation. See also comment on Figure 5.

This is a very interesting comment. As we understand it, what is meant is that presynaptic firing slows down, but facilitation somehow compensates by making larger EPSPs. How this would play out in postsynaptic firing however is not clear and would depend on that relationship between the time constant of presynaptic adaption determining EPSP rate and the postsynaptic membrane time constant which determines temporal integration.

line 405: As the IC is at least one synapse later than the VCN input, and has a longer pathway to travel (both ascending and descending), it would be expected that it is also delayed. In the experiments where two inputs are combined, was this delay taken into account?

A delay in IC input was not taken into account. Adding a delay would also mean assuming that these descending IC neurons solely respond to acoustically driven input—which is unknown.

line 373: "small number" : be more specific.

Added “(N = 3)”.

line 486: This needs a few citations. There are examples in the auditory system that may be relevant (or even lead to an opposing conclusion), but at least some backup for the statement should be present so readers know what you are thinking about.

Thank you for pointing this out. We added citations that were relevant to the efferent system.

Line 530: The speed of the receptor kinetics might not be the important observation here given the relatively slow output of the MOC system. Did you consider that the receptor-associated calcium influx might be important? This could be discussed.

We added to the discussion (see lines 555-557).

line 572: "Insertion" seems the wrong word here. Perhaps "Activation"?

Thank you. Done.

Line 670: this should include "uncompensated" in the description for clarity.

Thank you. Done.

line 673: "baselined" ? You mean the baseline current (define time frame) was subtracted from the data. Please reword.

Thank you. Done.

line 734-735: The tau values and amplitudes have units; I assume these are in sec and nA respectively, but they should be specified. The meaning of "n" and "n0" is not defined.

For these equations, tau and amplitude are unitless. This is because they are used as scaling factors, affecting the amplitude for each individual EPSG by weighting Gmax. Tau could alternatively be described as a stimulus constant, e.g. Tau = 12.9 stim #, but we believe this is still better expressed as a unitless value. We tried to clarify by specifying that the exponential equations were fit to normalized physiological data, and by simplifying the equations (n_0_ always equals 0, so n_0_ was redundant and therefore removed).

Figure 4Ai shows T-stellate cells projecting to the DCN in the schematic. However, 4Ei does not show any visible terminals in the DCN. Also, the lack of terminals in the DCN is not mentioned in the text (lines 216-218). At face value, this suggests that those T-stellate cells that project to the DCN are not the same population that projects to the IC. However, this would contradict data in the cat (Adams, 1983), which shows that double-labeling of a substantial population of "multipolar" cells in the VCN after combined IC and DCN injections. On the other hand, it is not clear from Oertel's reconstructions that all T-stellate cells have a collateral to the DCN (though clearly limitations on that data). Those of Smith and Rhode's (89) cat paper, where axons could be traced, do. Perhaps some comment on this would be appropriate.

Perhaps the reviewer was viewing a printed version of this figure? Terminals in the DCN are clearly visible in Figure 4Ei, and to a much lesser extent in Eii. The purple labeling in DCN is not background as it is absent in the DCN molecular layer. Our data agree with Adams, 1983, and Smith and Rhode’s, 1989—that at least a subset of, if not all, IC projecting T-stellate cells send collaterals to the DCN. See Figure 4Ei

Figure 5: The sample size for the loose-patch cell attached recordings in the IC is not given.

This is now given in the text.

Reviewer #3:In addition to the main conclusion, the manuscript includes a general description of MOC neuron physiology in mature mice using a reporter line to visualize cholinergic neurons. These descriptions could serve as markers to identify MOC neurons in future experiments even without the reporter line.Overall, the findings are very interesting as they question the MOC system as a reflex arc, given that the ascending input can be overwritten or at least modified by descending commands. The manuscript is prepared at the highest standard. The experiments are well constructed and of excellent quality. Especially the use of an intersectional AAV guided expression of channelrhodopsin to study the origin and function of MOC inputs and performing 2-electrode dynamic clamp to examine their interplay is very elegant.Including the role of the known inhibitory input to the MOC neurons into the evaluation of the MOC output would add an important piece to our understanding of this efferent circuit.I have only a few suggestions that could further improve the functional interpretation.The manuscript focusses strictly on glutamatergic activation of AMPA receptors. This raises two questions: First, do MOC cells express NMDA receptors or can we neglect them?

MOC neurons do seem to express NMDARs, but no obvious NMDAR mediated current was apparent during presynaptic stimulation of either ascending or descending input. We did observe NMDAR mediated currents during somatic puff application of 1 mM glutamate, but deliberately chose to focus on AMPARs for this study. It would be interesting if NMDARs play a role in plasticity in these neurons, but we were not able to induce LTP using our opsin approach, despite trying.

The second and major question is: How is inhibition integrated into generating the MOC output? The conductance clamp experiment is very good and I am not suggesting to do a 3-electrode recording, but it would be very useful to use the published (Torres Cadenas et al., 2020) information on the inhibitory input to the MOC cells to test their influence in the conductance clamp experiment. This could be done testing one excitatory input at a time in combination with the inhibition. Especially with respect to the apparent discrepancy between the high firing rates suggested by the current injection protocols and the rather low firing rates in vivo. As discussed by the authors, it could be the effect of anesthesia, but it could also be the impact of inhibition. Only then, a good prediction of MOC output should be possible.

A new set of experiments exploring the impact of inhibitory inputs will be important to do, but we feel are beyond the scope of this study. We added a reference to Torres Cadenas, 2020 and a discussion of the potential roles for inhibition (lines 483-498).

[Editors' note: further revisions were suggested prior to acceptance, as described below.]

The manuscript has been improved, but some remaining issues have been raised by reviewer #2 that need to be addressed, as outlined below:1. Regarding: "To these points we would like to note that the T-stellate cell firing in vivo does not mirror the adapting responses of the afferent neurons. Rather they show the characteristic 'chopper' responses described by Sachs and colleagues years ago. Moreover, we found that the chopper response of the T-stellate cell is recapitulated by simply applying current injection stimuli repeated over and over."I think that the authors and I have somewhat different readings of the past literature here. "Choppers" in the VCN exist as a continuum of discharge patterns in cat (Godfrey et al. 1976; Bourk, 1976; Typlt et al., 2012), some of which have pronounced adaptation during short tone bursts (Bourk, 1976; Young et al., J. Neurophys. 1988; Blackburn and Sachs, J. Neurophysiol. 1989; Winter and Palmer, Hearing Res. 1990 Figure 7; guinea pig; Rhode and Smith, 1986, Figure 17 at 40dB SPL). Similar adaptation is evident in mouse (see Roos and May, Hearing Res. 2012 Figures 5 and 6). Quantification of regularity in responses to longer tone bursts (seen in Godfrey et al., 1975) is generally lacking in the literature. The point is that in some "choppers", there is rate adaptation, driven by a combination of the multiple phases of adaptation in the auditory nerve rate, and by inhibition that increases over time. I agree that most stellate cells in the CN show little adaptation in slice recordings in response to direct current injection, as indicated by the authors, but a few do (Figure 3, Manis et al., 2019).It is ok for the authors to do the experiment in the way that they did, but they also should not ignore the literature that indicates that in responses to tone bursts in vivo, adaptation over the time scale of their stimulus sets does occur in some potential afferent cell populations.

Thank you. We have updated lines 367-379, and 615 to reflect that sustained firing in response to tones is exhibited by the majority of choppers, a proportion do indeed exhibit adaptation of different degrees indicating a continuum. Exploring the cellular bases of such types (input adaptation or growing inhibition) or the degree to which these different types drive MOC firing will be a fruitful direction for future work.

2. Regarding: "A second limitation is that the synaptic dynamics are treated as independent of frequency. This assumption conflicts with data shown in the paper, where the depression and facilitation rates are frequency-dependent in a manner typical of many previously studied synapses." Response: "We respectfully disagree with this point. Our figure 7D clearly shows profiles of depression or facilitation that are identical after nearly tripling the stimulation rate, from 20 to 50 Hz."This depends on the assumptions that are made regarding the dynamic behavior of synapses when challenged with different rates of spikes, and on the lack of data at these synapses regarding the dynamic behavior across a wider range of stimulus frequencies. The data shown in the paper at two frequencies are plotted as a function of spike number during the train, which translates into two different depression rates (e.g., time constants). Synapses are inherently non-linear, so that measurements made at two relatively low frequencies (20, 50 Hz) should not be extrapolated to higher frequencies (180 Hz) without acknowledging this limitation. The demonstration by Wang and Kaczmarek (1998) of the engagement of a frequency-dependent recovery from depression at high rates is a clear example of this point (and one that has been shown at other synapses). The authors should recognize in the text that they have used a simplified situation in which rate-dependent factors that may accrue outside the range that they were able to study are not considered. I do not think it much affects their conclusions as far as they go, but it is just a caveat worth mentioning.

Thank you for clarifying this point. We added lines 774-776 to make it more transparent that the short-term plasticity simulated in our conductance clamp experiments go beyond the presynaptic frequency range we were able to study, and that the exponential fits from our physiological data were extrapolated to higher rates.

3. The issue with dynamic clamp is a real one, but is also second-order. It also cannot be properly evaluated because the roles of dendritic ion channels in these cells are not known, and the consequences of dendritic conductance changes (e.g., synaptic inputs) and somatic conductance changes imposed by dynamic clamp are not exactly the same, although they are qualitatively very similar. I accept that you do the experiments that you can; however I do wish that the authors would acknowledge that there are limitations to this approach.

Thank you again for clarifying this point. We have added some text on lines 346-349 to acknowledge the limitation of somatic conductance clamp.